# PackQViT: Faster Sub-8-bit Vision Transformers via Full and Packed Quantization on the Mobile

**Peiyan Dong**[1]*, **Lei Lu**[1]*, **Chao Wu**[1]*, **Cheng Lyu**[1], **Geng Yuan**[2], **Hao Tang**[3]†, **Yanzhi Wang**[1]

[1]Northeastern University, [2]University of Georgia, [3]Carnegie Mellon University

{dong.pe, lu.lei1, cha.wu, yanz.wang}@northeastern.edu

## Abstract

While Vision Transformers (ViTs) have undoubtedly made impressive strides in computer vision (CV), their intricate network structures necessitate substantial computation and memory resources. A decision-making process for CV tasks typically entails performing computations with low latency, which is a tricky problem for ViT models. Model quantization is a widely-used technique to optimize the hardware efficiency of deep neural networks. Full quantization under Sub-8-bit precision, in particular, is a promising solution to reduce inference latency significantly. Unfortunately, current commodity hardware, such as CPUs and GPUs, still struggles to efficiently execute these sub-8-bit quantized networks, as their SIMD instructions only support a granularity of 8 bits or wider. Also, there is a scarcity of literature that presents a full quantization paradigm for ViTs. In this paper, we propose an activation-aware fully sub-8-bit quantization-aware training (QAT) framework called **PackQViT** for efficient yet accurate ViT acceleration on mobile devices to facilitate real-time AI-powered decision-making. Specifically, in revisiting data activation within the ViT dataflow, two characteristics are relevant to quantization strategy and precision: the long-tailed distribution and systematic channel-wise outliers. In response, we employ either *log2 quantization or clipping* to address the long-tailed distribution and incorporate *outlier-aware training* for residual link quantization to regulate the various channel-wise outliers more consistently. Notably, due to the systematic fixed pattern, *outlier-aware training* approach can predict the channel indices and regularized scales of outliers in advance, thus avoiding the runtime data-adaptive selection during inference. Furthermore, we employ Int-$2^n$-Softmax, Int-LayerNorm, and Integer GELU to enable *integer-only* computation flow. Finally, we develop a *SIMD-based 4-bit packed multiplier* to achieve end-to-end ViT acceleration on mobile phones. Compared to prior studies on ViT quantization using 8-bit precision, PackQViT surpasses other works by an improved accuracy ranging from 0.4% to 17.9% for various widely used ViTs on ImageNet dataset; under 4-bit precision, PackQViT demonstrates 0.4%~2.8% higher accuracy. Compared to the baseline multiplier, our implementations on the Realme GT Android smartphone with Snapdragon 870 SoC CPU achieve $2.6\times \sim 3.7\times$ speedup under 8-bit scenario and $3.8\times \sim 5.9\times$ speedup under 4-bit which ensures practical real-time performance. Codes available at PackQViT.

## 1 Introduction

Transformers [3, 35] have experienced a resurgence in recent times, with ViTs [16] demonstrating remarkable versatility across a broad range of domains, including computer vision (CV), e.g., image

---

*Equal Contribution

†Corresponding Author

37th Conference on Neural Information Processing Systems (NeurIPS 2023).

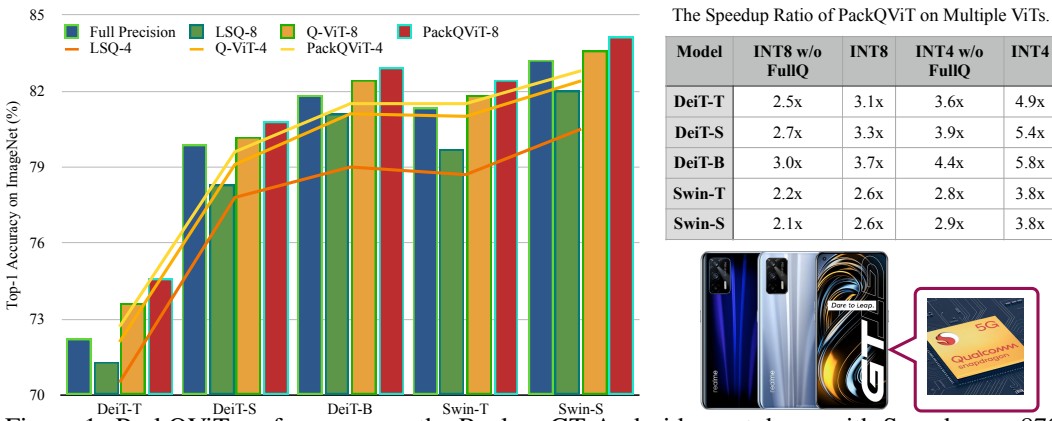

The Speedup Ratio of PackQViT on Multiple ViTs.

| Model | INT8 w/o FullQ | INT8 | INT4 w/o FullQ | INT4 |
|---|---|---|---|---|
| DeiT-T | 2.5x | 3.1x | 3.6x | 4.9x |
| DeiT-S | 2.7x | 3.3x | 3.9x | 5.4x |
| DeiT-B | 3.0x | 3.7x | 4.4x | 5.8x |
| Swin-T | 2.2x | 2.6x | 2.8x | 3.8x |
| Swin-S | 2.1x | 2.6x | 2.9x | 3.8x |

Figure 1: PackQViT performance on the Realme GT Android smartphone with Snapdragon 870 SoC CPU. Left: Top-1 accuracy (%) on ImageNet compared to SOTA QAT methods on 8-bit/4-bit precision. Right: The speedup ratio (x) compared to FP32 inference.

classification [16], object detection [9, 53], semantic segmentation [51], image processing [10], and video understanding [52], as well as in complex scenarios that involve multi-modal data. Moreover, ViTs have the potential to unify diverse application domains through shared architectures, addressing two of the most pressing challenges in deep learning: ● the reliance on limited domain-specific data; ● the need for constant model refinement to meet evolving demands. Given these strengths, ViTs are poised to become a dominant force in the field of deep learning.

On the downside, ViTs are usually times slower than competitive convolutional neural networks (CNNs). Many factors limit the inference speed of ViT, including the massive number of parameters and quadratic-increasing computation complexity with respect to token length. Model quantization is a promising approach to address the above-mentioned issues. Still, it faces the following challenges before ViTs become an indispensable staple of real-world applications on resource-constrained hardware (e.g., augmented or virtual reality applications on mobile devices). (i) Researchers [23, 27, 11, 44, 34, 50, 36] have developed quantization techniques to reduce both computation and communication requirements for Transformers. However, most of these techniques are based on training-free Post-Quantization (PTQ), which was originally proposed for natural language processing (NLP) oriented Transformers [49, 14, 38, 6, 48, 46] and may lead to accuracy drops when directly applied to ViTs. NLP-oriented transformers tend to have a larger model size than ViT [14], making it difficult to utilize QAT on limited computing resources. In contrast, ViT can adapt QAT to minimize the quantization error incurred in PTQ. Although QAT has been applied to ViT in [29], it lacked an analysis of data distribution within networks and was not fully quantized, which constrained task accuracy and practical implementation. (ii) To enhance accuracy, ViTs use more hardware-unfriendly computations than CNNs (e.g., GELU, LayerNorm, and Softmax). Previous methods [34, 29] do not quantize Softmax/LayerNorm/GELU because quantization may cause significant accuracy degradation. However, data moving between different data domains, e.g., dequantizing and requantizing, and data movement between floating-point and integer domains, will cause more hardware overhead. (iii) CPUs utilize SIMD units to perform multiple operations in parallel efficiently. SIMD instructions can effectively exploit byte-level data (8-bit integers) parallelism and are well supported in mainstream ISAs and DNN processing frameworks, e.g., GEMMLOWP [24] in TensorFlow-Lite, and QNNPACK [18] in PyTorch. However, these low-precision libraries are ineffective for running sub-8-bit (bit-width<8) quantized networks, as the SIMD-based units only support data parallel execution for 8 bits or wider. Consequently, this poses challenges in implementing sub-8-bit ViTs on mobile with both acceptable accuracy and lower runtime speed. In short, we should address the hardware implementation issue while enjoying the additional optimization dimension provided by multi-head self-attention.

In this paper, we present a novel framework called PackQViT for efficient and accurate end-to-end ViT inference on mobile devices using activation-aware full sub-8-bit quantization. Our approach involves conducting hardware profiling of ViTs on commercial mobile CPUs to identify an end-to-end acceleration scheme and model full quantization. To decrease data precision without compromising model accuracy, we revisit the data distribution in ViTs. Weight data has a typical normal distribution, while activation data has the long-tailed distribution (attention maps and activation after GELU) and systematic channel-wise outliers in the addition of residual links. Thus, we implement uniform

quantization on weight, log2 quantization on attention maps, and uniform quantization on activation values after GELU with the clipped range [-1, 10]. To address the issue of channel-wise outliers, we introduce an outlier-aware training for the addition of residual links. The outlier-aware training can predict the channel indices and regularize scales of outliers with a power-of-two ratio, avoiding the complex hardware control logic for dynamic data selection during runtime inference. We modify the inference formulas of Softmax, LayerNorm, and GELU, enabling integer-only inference. Finally, we develop a SIMD-based 4-bit packed multiplier to accelerate ViT inference on mobile devices.

Compared to state-of-the-art (SOTA) ViT quantization studies [23, 27, 11, 44, 34] as shown in Figure 1, PackQViT achieves better accuracy: Under 8-bit precision, PackQViT can achieve 0.4%∼5.8% higher accuracy than other works and 0.9%∼2.3% better accuracy than the full precision models; PackQViT-Swin can achieve 1.4%∼17.9% higher accuracy than other works and 0.9% better accuracy than the full precision models. Under 4-bit precision, PackQViT can achieve 0.4%∼2.8% higher accuracy than others and <0.4% accuracy drops than full precision models. Compared to the baseline multiplier on the Realme GT Android smartphone, our inference acceleration PackQViT achieves $2.6\times \sim 3.7\times$ speedup (8-bit) and $3.8\times \sim 5.9\times$ speedup (4-bit). According to our knowledge, PackQViT reaches the real-time performance (34.8 ms in DeiT-T) of classic ViTs on the mobile phone for the first time. Overall, our contributions are summarized as follows:

- We propose an activation-aware sub-8-bit full QAT framework for the efficient and effective inference of ViTs.
- We propose an end-to-end integer framework equipped with a SIMD-based 4-bit packed multiplier for real-time ViT inference on mobile devices.
- We conduct experiments to showcase the superior inference accuracy of PackQViT under the sub-8-bit scenario compared to state-of-the-art ViT quantization studies, while also highlighting its significant hardware efficiency.

## 2 Background and Related Work

### 2.1 Vision Transformers

Transformers which consist of a multi-head self-attention (MSA) module and an MLP (FFN) module, both with LN at the beginning, were initially designed to tackle long sequence learning in NLP tasks. However, after the impressive success of a pure transformer architecture for image classification [17], the interest in transformers for CV surged. This universality of transformer architectures from NLP to CV is attributed to the more uniform representations across all layers than CNNs, self-attention mechanism enabling early aggregation of global information, and ViT residual connections that propagate features strongly from lower to higher layers [37]. Consequently, several ViTs [4, 40, 1, 21, 39, 47, 45, 28] have been proposed for various CV tasks, including object detection, semantic segmentation, and image retrieval, achieving competitive performance against CNN counterparts.

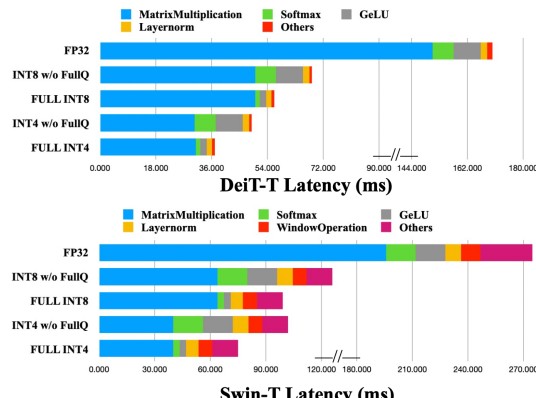

Figure 2: DeiT-T/Swin-T breakdown on Realme GT Android smartphone CPU.

### 2.2 End-to-End Acceleration Methods

**Hardware Profiling Analysis.** To better understand the runtime breakdown for ViTs, we profile DeiT-T and Swin-T, popular ViT models, on the Snapdragon 870 on-board CPU. We present the model in FP32 precision, INT4 w/o full quantization, and INT4 w/ full quantization. And non-full quantization means LayerNorm/Softmax/GELU are still in the floating-point domain. In Figure 2, we observe that for the full precision DeiT-T on an Android smartphone with a Snapdragon 870 SoC CPU, matrix multiplications, i.e., Linear Transform, $Q*K^T$, Attention*V, Linear Projection, FC1, FC2, occupy the latency distribution close to 88.51%; Nonlinear operations, i.e., GELU, LayerNorm, SoftMax, merely

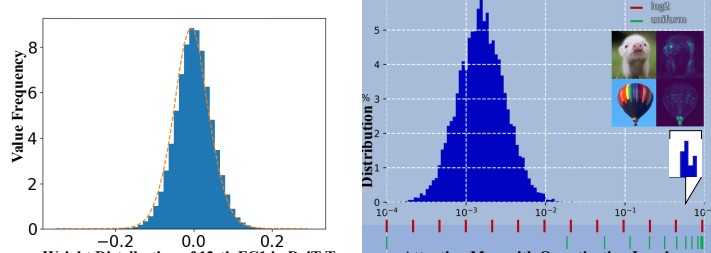
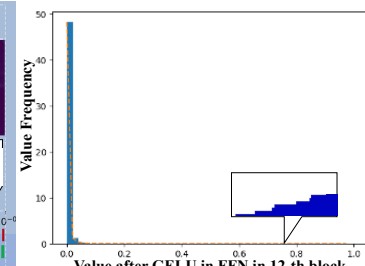

Figure 3: Left: A representative normal distribution of the weight of the 12-th FC1 layer in DeiT-T. Middle and Right: Long-tail distribution for attention map and activation after GELU.

take the 10.16% of latency distribution. However, under the 4-bit w/o full quantization, nonlinear operations take 44.61% of latency distribution. Suppose we do the full 4-bit quantization on the model. In that case, nonlinear operations will only take about 24.21% of latency distribution, and the model speedup ratio will be improved from 3.6x to 4.9x as Table in Feature 1. Similar to DeiT-T, the speedup ratio of Swin-T will be increased from 2.8x to 3.8x. Hence, research on full quantization for end-to-end acceleration is imperative.

**Quantization on Transformers.** Quantization is one of the most powerful ways to decrease neural networks' computational time and memory consumption. It uses low-bit representations for weight and activation tensors. Low-bit fixed-point representations, such as INT8 and INT4, further reduce energy consumption as the fixed-point operations are more efficient than their floating-point counterparts [22]. Current quantization methods can be divided into two categories: QAT and PTQ. NLP-oriented Transformers mainly employ PTQ for three reasons [49, 14, 48]: • too large model size (usually over 350M). • not accessible dataset. • limited academic computational resources to support the training of large language models. However, ViT's small model size and availability of public datasets make it suitable for QAT, which avoids the issues associated with PTQ that can lead to sub-optimal model performance or significant accuracy reduction without fine-tuning. To illustrate, previous PTQ approaches [34, 31] quantized ViT models to 8 bits, resulting in only a 1.2%~1.8% accuracy decrease. [29] proposes a QAT method for ViTs with information-rectified and similarity-aware strategies. However, this work does not quantize Softmax, LayerNorm, and GELU modules, which hold significant execution time, e.g., 44.61% of latency distribution for 4-bit DeiT-T execution and 42.74% for 4-bit Swin-T execution on mobile CPU (Figure 2). In PackQViT, we aim to implement an accurate, fully quantized ViT under QAT.

## 2.3 Low-Precision Linear Algebra Kernels

Low-precision linear algebra kernels aim to maximize computing throughput on low-precision operands by extending existing wider bit-width linear algebra kernels. Using lower-precision operands has been shown to improve performance in two ways. i) caches to fit more data and ii) lower-precision SIMD instructions to be utilized (e.g., vmlaq s8() in ARMv8 ISA) to process more elements in parallel than higher-precision instructions (e.g., vmlaq f32()). SOTA low-precision linear algebra kernels, such as Google's GEMMLOWP [24] and Facebook's QNNPACK [18], have been developed to maximize the throughput of low-precision operands, which are highly effective in improving the DNN inference efficiency under the 8-bit quantization (W8A8) scenario, e.g., 3× end-to-end speedup compared to the FP32 baseline when running on PyTorch with a 64-bit ARM Cortex-A72 CPU [2]. However, for more aggressive sub-8-bit quantization, it does not provide additional performance benefits because commodity CPUs only support Single Instruction Multiple Data (SIMD) operations of 8-bit or greater precision. As a result, low-precision kernels merely zero-extend the sub-8-bit operands to align them with byte boundaries, treating them as 8-bit operands.

## 3 Revisiting Data Distribution within ViTs

Unlike [32] which merely analyzes the activation value in the attention map and LayerNorm, we conduct a comprehensive analysis of the data distribution (weight & activation) of ViTs. • **Weight.** The data has a standard normal distribution as shown in Figure 3. • **Activation.** Two characteristics affect the quantization strategy: *long-tail distribution* – attention maps & activation after GELU and *channel-wise outliers* – in the addition of the residual link.

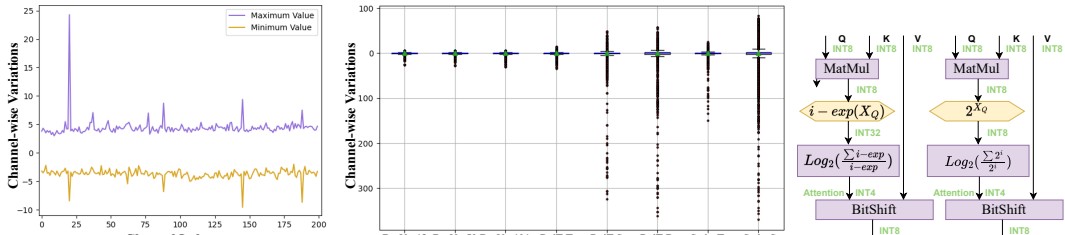

Figure 4: Left: Channel-wise minimum and maximum values of the second residual link addition in the $9^{th}$ block of Swin-T. Middle: Channel-wise ranges of the last residual link addition in representative models. Right: Comparison of common Int-Softmax [32] and Int-$2^n$-Softmax in quantized MSA inference.

**Long-Tail Distribution.** • Attention map. As model sizes increase, the storage and computation required for the Softmax with an attention map in each head create a bottleneck that substantially impacts the throughput and latency of inference (see Section 2.2). To address this issue, we also investigated attention maps, which reveal a long-tail distribution (see Figure 3). Based on token pruning techniques [26, 15], the image-level redundancy causes most of the attention values to be centered around a small value and the rest of the values are discrete and converge to 1 as Figure 3. Two cases are put here to show the attention sparsity. Compared to the uniform quantization which assigns only one bin to such a large number of values, the log2 method (only with 4-bit) has more resolution to cover this data range. • Activation after GELU. The GELU activation function in FFN leads to a truncation effect that heavily concentrates the resulting values around zero, showcasing a clear one-sided stacking pattern (see Figure 3(b)).

**Channel-Wise Outliers.** • A serious inter-channel variation in the addition of residual link. Figure 4 depicts the channel-wise outliers in the last residual link. To provide a comparison, we display the activations of ResNets in the last outputs of the 4th step. Notably, the channel-wise ranges in ViTs exhibit more significant fluctuations than in ResNets. This is because the data flow of nonlinear functions, such as Softmax and GELU, existing in the residual link of transformer structures, results in a greater disparity between the input and output of the residual blocks compared to CNN models. These results suggest that layer-wise quantization, with the same quantization parameters for all channels, would cause an unacceptable quantization error. • Systematic outliers. Although outliers basically appear in every sequence, they are concentrated in $< 6\%$ *fixed* feature dimensions of the *fixed* encoder blocks, as shown in Figure 4 (Left) (see Appendix 8). For instance, when testing 1024 images, outliers always occurred in the $21^{st}$, $89^{th}$, $146^{th}$, and $189^{th}$ channel of the $9^{th}$ block in Swin-T. Furthermore, setting these outlier features to zero in DeiT-T causes the top-1 accuracy drop by 30% on ImageNet classification. Since the outlier locations follow a *fixed* pattern, we can anticipate them in advance, obviating the hardware design with complex control logic to support online data-adaptive selection.

## 4 Activation-Aware Fully-Quantized ViTs

In detail, Section 4.1 presents the preliminary of quantization. In Sections 4.2-4.3, we propose two novel quantization methods to fit the distinct activation distribution inside ViTs, *long-tail-aware quantization*, and *outlier-aware training* for *systematic channel-wise outliers* quantization. Section 4.4 shows the *SIMD-based 4-bit packed multiplier* to support the 4-bit computation on mobiles practically.

### 4.1 Preliminary

Given the quantization bit-width is $b$, the quantizer $Q(X|b)$ can be formulated as a function mapping a floating-point number $X \in R$ to the nearest quantization bin. Among various quantizers, uniform [23] and log2 [7] are generally used. Apart from the special data distribution mentioned in Section 3, we apply symmetric layer-wise uniform quantization on weights and asymmetric channel-wise uniform quantization on activations. For QAT training, the forward and backpropagation of the quantization

function in a quantized network is formulated as

$$\text{Forward}: Q-Linear(x) = \widehat{x} \cdot \widehat{w} = s_x s_w (Q_X(x) + \frac{z}{s_x}) \cdot Q_w(w),$$

$$\text{Backward}: \frac{\partial \zeta}{\partial x} = \frac{\partial \zeta}{\partial \widehat{x}} \frac{\partial \widehat{x}}{\partial x} = \begin{cases} \frac{\partial \zeta}{\partial \widehat{x}} & \text{if } x \in [l_x, u_x] \\ 0 & \text{otherwise} \end{cases},$$

$$\frac{\partial \zeta}{\partial w} = \frac{\partial \zeta}{\partial x} \frac{\partial x}{\partial \widehat{w}} \frac{\partial \widehat{w}}{\partial w} = \begin{cases} \frac{\partial \zeta}{\partial x} \frac{\partial x}{\partial \widehat{w}} & \text{if } w \in [l_w, u_w] \\ 0 & \text{otherwise} \end{cases}, \tag{1}$$

$\zeta$ is loss function, straight-through estimator [5] is used to obtain the derivation of the gradient in backpropagation for $Q(\cdot)$. To achieve a fully quantized ViT, we leverage the analysis in Section 3 and quantize all modules, including Conv, Linear, MatMul, LayerNorm, Softmax, GELU, etc.

### 4.2 Long-Tail-Aware Quantization

Two activation feature maps in dataflow exhibit a long-tail distribution: the *attention map* and the *activation after GELU*. GELU truncates activation values to be concentrated around 0, while the rest keep the original normal distribution. To allocate enough precision bits, we *clip* the activation value to [-1, 10] after GELU with uniform quantization. Moreover, we adapt GELU to support integer-only inference and regularize the quantization error as [15]. In this context, we focus on the attention map.

**Log2 Quantization for Attention Map.** Since the output range $(0, 1)$ of Softmax makes the log2 quantization [7] calibration-free and log2 can convert the MatMul to more hardware-efficient BitShift, we propose:

$$\text{Attn}_Q = Q(\text{Attn}|b) = \text{clip}(\lfloor -\log_2(\text{Attn}) \rceil, 0, 2^b - 1), \tag{2}$$

$$\text{Attn}_Q \cdot V_Q = 2^{-\text{Attn}_Q} \cdot V_Q = V_Q >> \text{Attn}_Q = \frac{1}{2^M} \cdot (V_Q << (M - \text{Attn}_Q)), \tag{3}$$

where $M = 2^b - 1$. Note that directly right shifting $V_Q$ with the $\text{Attn}_Q$ may result in severe truncation error. $M - \text{Attn}_Q$ with scaling $\frac{1}{2^M}$ can prevent the error through a left-shift operation.

**Inference with Int-$2^n$-Softmax.** Replacing the natural constant $e$ inside the Softmax with 2 [8] as shown in Figure 3, we propose Int-$2^n$-Softmax with Log2 inference:

$$\exp(s \cdot X_Q) \approx s' \cdot 2^{X_Q}, \quad \text{Int-Softmax}(s \cdot X_Q) = M - \log_2 \lfloor \frac{\sum 2^{X_Q}}{2^{X_Q}} \rceil. \tag{4}$$

BitShift is implemented for the $2^{X_Q}$ and also for the integer log2 function. Experiments show that the replacement does not cause an accuracy drop under the same training receipt [42] (see Appendix 8).

We illustrate the difference between normal Int-Softmax [25] and our method in Figure 4. On the left, we present the common Int-Softmax with log2 quantization. On the right, our Int-$2^n$-Softmax replaces the floating-point exponential calculation with BitShift. Notably, i-exp is a second-order polynomial that still involves the multiplication and addition in the floating-point domain, while Int-$2^n$-Softmax employs a 4-bit representation on attention maps and replaces multiplication with BitShift operations, thereby enabling the inter-only dataflow, and reducing both computation and memory footprint.

### 4.3 Outlier-Aware Training for Systematic Channel-Wise Outliers Quantization

The addition of residual links contains many systematic and channel-wise outliers. To mitigate this, we propose the *outlier-aware training* approach that predicts the precise channel indices of the addition of residual links and regulated scales of the outliers with the *power-of-two ratio (PTR)*. Thanks to the systematic outliers, outlier-aware training can obviate the need for complex control logic during inference to support data-adaptive selection.

**Power-of-Two Ratio for Residual Link Quantization.** Given the input activation $X \in B \times L \times C$, and the PTR $r \in N^C$, then the quantized activation $X_Q$ can be formulated as:

$$X_Q = Q(X|b) = \text{clip}(\lfloor \frac{X}{2^r s} \rceil + z, 0, 2^b - 1), \quad s = \frac{\max(X) - \min(X)}{2^R(2^b - 1)}, \quad z = \text{clip}(\lfloor -\frac{\min(X)}{\max(X)} \rceil, 0, 2^b - 1). \tag{5}$$

**Algorithm 1:** Outlier-Aware Training

1   Given the full-precision ViT Model, the test subdataset of target Dataset $D$, the number of encoder blocks $L$, $Epoch_s$ for searching, $Epoch_f$ for fine-tuning, and quantized low-bit $b$;

    `// Step1: Initialize the PTC `$r$` with the outlier estimated from the full-precision Model.`

2   **foreach** $l \in [0, 1, \ldots, L-1]$ **do**

3      $i_o, r_o = \text{Check\_Outliers}_{3\sigma}(\text{Model}_l, \text{D})$;

4      Quantize $\text{Model}_l$ by Eq. (5) with $i_o, r_o, b$ into the $\text{QModel}_l$;

5   **end**

    `// Step2: Search for the channel index of outliers and determine the PTR `$r$` by the l2 regularization.`

6   $r = r_o, i = i_o$;

7   **foreach** $eps \in [0, 1, \ldots, Epoch_s - 1]$ **do**

    `// These two operations are gradient-free.`

8      $i, r = \text{Check\_Outliers}_{3\sigma}(\text{Model}_l)$;

9      $r_i = \underset{r_i \in \{1,2,\ldots,R\}}{argmin} \; ||X_i - \lfloor \frac{X_i}{2^{r_i}s} \rceil \cdot 2^{r_i}s||_2$ ;

    `// Following Eq. (2), Eq. (4) and Eq. (5).`

10     $\text{task}_{loss}, \text{quantization}_{loss} = \text{Quantize}(\text{QModel}, b)$;

11     $\text{Backward}(\text{task}_{loss}, \text{quantization}_{loss})$;

12   **end**

    `// Step3: Finetune the `$b$`-bit quantizaton.`

13   Fix $r$ and $i$ for outliers and quantize QModel with fine-tune $Epoch_f$;

14   Finalize the quantized ViT Model.

---

Hyperparameter $R=max(r)=4$ satisfies different inter-channel variations across models. We predict each outlier's channel index $i$ and PTR $r$ by the following outlier-aware training algorithm.

**Outlier-Aware Training.** This algorithm is divided into three steps: ● Initialize the PTR by detecting the outlier of the full-precision model with $3\sigma$ method [12]. ● Search for the channel index $i$ and the PTR $r$ of each outlier with the $l_2$ minimization. ● Fix the index $i$ and $r$ obtained in step 2 and fine-tune the model (see Algorithm 1).

**Inference with Int-LayerNorm.** During inference, with the gathered $i$ and $r$ of systematic outliers, we extract the layer-wise parameters $s$ and $z$, and compute $\mu$, $\sigma$ in the integer domain. Meanwhile, thanks to the BitShift, PTR $r$ can be calculated with the quantized Layernorm efficiently (also refer to [25] and see Appendix 8).

### 4.4 SIMD-Based 4-Bit Packed Multiplier

In the 8-bit multiplier of the SIMD CPU, INT4 data is represented as INT8 with the high 4 bits set to 0, which means there is no benefit for the efficiency improvement of INT4 computation. We develop our *SIMD-based 4-bit packed multiplier* (Figure 6): 1) Dot-multiplication. In the SIMD kernel, we concatenate two weights $W_{i,j}$ and $W_{i+1,j}$ from adjacent rows and multiply them with their shared activation value. The output is an INT16 data type, with the first 8 bits representing the result of the multiplication with $W_{i,j}$, and the last 8 bits corresponding to $W_{i+1,j}$. Following the SIMD memory mechanism where 16-bit of memory footprint will be freed up for the result of each dot-multiplication, we can avoid potential overflow issues. 2) Addition. Combined with the Bitshift operator, expand the 16-bit output in Step 1 into 32-bit, where the $1^{st}$ 8 bits are 0, the $2^{nd}$ 8 bits containing $\text{Output}_{i,j}$, the $3^{rd}$ being 0, and 4th containing $\text{Output}_{i+1,j}$. Next, we perform a row-by-row summation. Since 32-bit of memory footprint will be supplied for each addition, this process can handle up to $2^8$ times additions without overflow, which is sufficient for multi-head attention (head-dimension = 32/64). Finally, we split the output into two INT16 values and quantize them back to INT4 in value-level, which can be fused into the GeMM kernel.

Our method is different from per-channel quantization for outliers. We perform layer-wise quantization on the activation matrix; all the channels share the same quantization parameters, i.e., scaling factors and zero-point. However, the predicted power-of-two coefficients will refine the outlier channels' scaling factors. In the practical implementation, those power-of-two coefficients will be equivalently mathematically transformed over the corresponding weights, as shown in Figure 5.

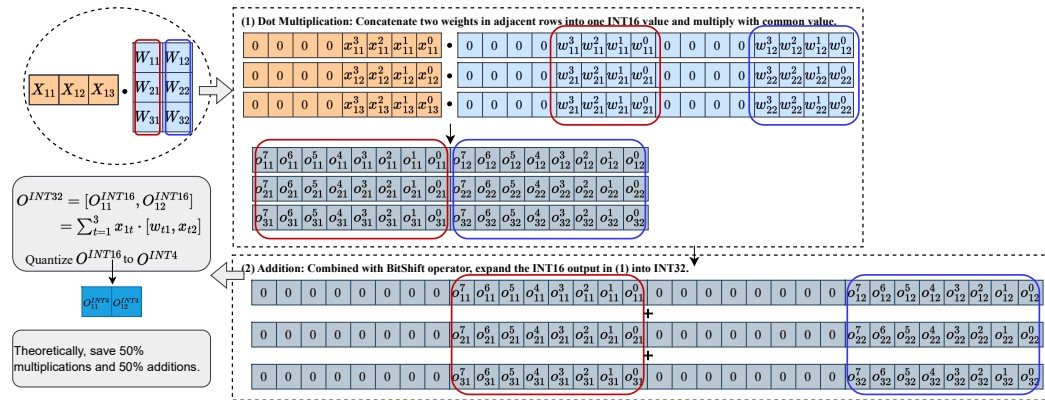

Figure 6: The paradigm of SIMD-based INT4 packed multiplier.

Our framework is compatible with per-channel quantization. However, common inference engines usually provide support only for layer-wise quantization on activation and per-channel quantization on weights, such as the Gemm and Convolution operator configuration. For example, ArmComputeLibrary [41] only supports channel-wise quantization configuration for weight matrix instead of the input activation.

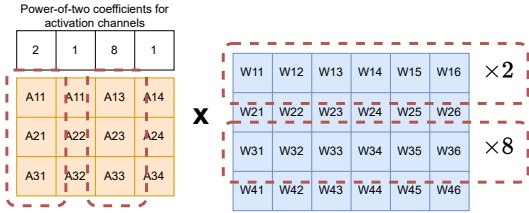

Figure 5: Matrix Multiplication Implementation for Power-of-Two Ratio.

## 5 Evaluation

Our experiments include activation-aware quantization and hardware implementation for multiple ViTs. To the best of our knowledge, there is no full quantization-aware training of ViTs at this point, so we reproduce and implement the Q-ViT [29] and LSQ [19] methods by ourselves. Based on the proposed hardware design logic of the 4-bit multiplier, our team is committed to enhancing the support for lower-bit precision, such as 3-bit and 2-bit, which will be a major area of our future development. Currently, we mainly present our model performance on 8-bit and 4-bit precision.

### 5.1 Experiment Setup

**Training Setup for PackQViT Framework.** The baseline models with 32-bit floating-point precision are from the TorchVision library [20]. Our experiments are conducted on the ImageNet-1K [13] with two popular vision transformer implementations, including DeiT [43] and Swin [33]. Apart from special notes, we apply symmetric layer-wise quantization on weights and asymmetric channel-wise quantization on activations. The hyperparameter $R$ in Power-of-Two Coefficients is set to 4. The training settings follow Q-ViT [29] with its DGD distillation, and the training process is executed on 4 NVIDIA V100 GPUs. Because we execute the outlier-aware training in a fine-tuned way, i.e., 70 epochs for step 2, 30 epochs for step 1, the training hours of the entire pipeline can be 37.5%~54% of train-from-scratch of full precision models (see Appendix 8).

**Hardware Platform.** To validate the practical deployment efficiency of the proposed PackQViT, we choose the Realme GT Master Android smartphone as the mobile platform, which is equipped with a Snapdragon 870 SoC consisting of the onboard Octa-core Kryo 585 CPU. We control the testing process to use a single thread and let the 3.2GHz Cortex-A77 Main Core perform the computation to obtain the most stable acceleration ratio with the proposed framework. The latency has been reported via 100 iterations for each test. Our inference engine is modified based on the ARM Compute Library. For the operators that are not yet supported in DeiT and Swin (such as Int-LayerNorm, Integer GELU, Int-$2^n$-Softmax), we implement them based on the design hierarchy of 'kernel' to 'operator' to 'runtime' function, using parallel optimization based on ARM processor's SIMD support.

### 5.2 Model Accuracy and Speed Performance

This paper employs several popular post-training quantization methods, including MinMax, EMA [23], Percentile [27], OMSE [11], Bit-Split [44], PTQ for ViT [34], and FQ-ViT [32]. Also, we compare with the SOTA quantization-aware training method, Q-ViT [29]. For the sake of fairness, we reproduced the results of Q-ViT with quantized LayerNorm and Softmax.

Table 1: Comparison of the Top-1 (%) accuracy with state-of-the-art methods on ImageNet dataset. We report bit width in the order of weight/activation/attention.

| Method | #Bits | DeiT-T [43] | DeiT-S [43] | DeiT-B [43] | Swin-T [33] | Swin-S [33] |
|---|---|---|---|---|---|---|
| Full Precision | 32/32/32 | 72.21 | 79.85 | 81.85 | 81.35 | 83.2 |
| PTQ | | | | | | |
| MinMax | 8/8/8 | 70.94 | 75.05 | 78.02 | 64.38 | 74.37 |
| EMA | 8/8/8 | 71.17 | 75.71 | 78.82 | 70.81 | 75.05 |
| Percentile | 8/8/8 | 71.47 | 76.57 | 78.37 | 78.78 | 78.12 |
| OMSE | 8/8/8 | 71.3 | 75.03 | 79.57 | 79.3 | 78.96 |
| Bit-Split | 8/8/8 | – | 77.06 | 79.42 | – | – |
| PTQ for ViT | 8/8/8 | – | 77.47 | 80.48 | – | – |
| FQ-ViT | 8/8/8 | 71.61 | 79.17 | 81.2 | 80.51 | 82.71 |
| QAT | | | | | | |
| Q-ViT | 8/8/8 | 73.6 | 80.2 | 82.4 | 81.8 | 83.6 |
| PackQViT | 8/8/8 | **74.6** | 80.8 | 82.9 | **82.4** | 84.1 |
| PackQViT | 8/8/4 | 74.5 | **80.8** | **82.9** | 82.3 | 84.1 |
| LSQ | 4/4/4 | 70.5 | 77.8 | 79 | 78.7 | 80.5 |
| Q-ViT | 4/4/4 | 72.1 | 79.1 | 81.1 | 81 | 82.4 |
| PackQViT | 4/4/4 | **72.7** | **79.6** | **81.5** | **81.5** | **82.8** |

Table 2: Comparision of Objection Detection using DETR-50 on COCO val2017.

| Methods | #Bits | mAP | AP50 | AP75 |
|---|---|---|---|---|
| FP32 | 32 | 59.5 | 83.3 | 64.7 |
| VT-PTQ | 8 | 57.6 | 82.3 | 63.1 |
| PackQViT | 8 | 58.3 | 82.9 | 63.9 |
| PackQViT | 4 | 55.6 | 82.2 | 60 |

Table 3: Evaluating the components on DeiT-T.

| Method | #Bits | Top-1 | #Bits | Top-1 |
|---|---|---|---|---|
| Full Precision | 32-32 | 72.2 | 32-32 | 72.2 |
| Baseline [29] | 8-8 | 73.6 | 4-4 | 72.1 |
| + Log2-Atten (Sec. 4.2) | 8-8 | 74.1 | 4-4 | 72.4 |
| + Outlier-aware Train (Sec. 4.3) | 8-8 | 74.2 | 4-4 | 72.5 |
| + Int-$2^4$-Softmax (Sec. 4.2) | 8-8 | 73.5 | 4-4 | 72.0 |
| + Int-LayerNorm (Sec. 4.3) | 8-8 | 73.6 | 4-4 | 72.1 |
| PackQViT | 8-8 | 74.5 | 4-4 | 72.7 |

**Image Classification on ImageNet.** In the 8-bit scenario, SOTA methods for PTQ have suffered a significant accuracy drop, ranging from 0.5% to 16.9% compared to the full precision baseline. However, QAT has shown promise in enhancing task accuracy by 0.5% to 2.4% over the baseline, thanks to its ability to minimize quantization errors and eliminate model redundancy. This suggests that redundancy in the original model hinders it from converging to the optimum, as confirmed by the quantized DeiT-T model achieving 1.4% to 2.4% higher accuracy than the baseline. While the current SOTA QAT method, Q-ViT, has made significant strides in correcting information distribution and knowledge transfer within ViTs, it falls short in analyzing the internal data flow, leaving room for improvement in quantization accuracy. Additionally, Q-ViT still relies on floating-point computations for Softmax, LayerNorm, and GELU, which makes it challenging for efficient hardware deployment. In contrast, our proposed PackQViT leverages activation flow fitting to achieve an additional accuracy boost of 0.6% to 1% over Q-ViT. Moreover, when switching the data precision of the attention matrix, the accuracy only reduces by less than 0.1%, indicating that a 4-bit log2 data type is sufficient to cover the data representation.

In the 4-bit scenario, PackQViT can effectively prevent accuracy degradation by less than 0.4%. Compared to other QAT methods, PackQViT still achieves a better accuracy performance of 0.4% to 2.8%. Furthermore, our method can be practically implemented on mobile devices using the ARM 8-bit multiplier and our proposed SIMD-based 4-bit Packed Multiplier.

**Object Detection on MSCOCO.** We compare object detection with transformers (DETR) [9], an end-to-end detector via a transformer encoder-decoder. We perform our quantization on DETR-R50. We compare the large-scale COCO dataset [30] as shown in Table 2. We compare our method under the 4-bit precision. We report the detection performance of the 8-bit PTQ method, VT-PTQ [34].

For this more complex task, our method has a negligible detection performance degradation by 1.2% mAP, which is better than the VT-PTQ method by 0.7 mAP. For more aggressive quantization under 4-bit, the model has 3.9 mAP performance drops. For more complex tasks, quantization tends to lead to a degradation of task performance.

**Latency Analysis.** Based on Table 4, we can draw the following conclusions: 8-bit quantization can bring an overall acceleration ratio of 2.5x to 3x depending on the size of the model, as the high computational workload on mobile devices can benefit from the higher utilization efficiency of limited memory resources under quantization algorithms. Specifically, INT8 quantization can achieve approximately 3.5x acceleration compared to FP32 in GeMM, which is very beneficial for models like Transformers, where most mathematical operations are matrix operations (>80%

Table 4: Latency (ms) on mainstream processors on edge platforms (1-thread).

| Model | Method | #Bits | Size(MB) | Android CPU (s) | RaspberryPi (s) | RISC-V (s) |
|---|---|---|---|---|---|---|
| DeiT-T | Full-precision | 32 | 20.1 | 0.172 | 23.97 | 3.4 |
| | PackQViT | 8 | 5.2 | 0.056 | 7.73 | 1.11 |
| | PackQViT | 4 | 2.7 | 0.037 | 5.15 | 0.74 |
| DeiT-S | Full-precision | 32 | 88.2 | 0.363 | 59.75 | 7.2 |
| | PackQViT | 8 | 22.2 | 0.11 | 17.5 | 2.2 |
| | PackQViT | 4 | 11.4 | 0.073 | 11.97 | 1.46 |
| DeiT-B | Full-precision | 32 | 346.2 | 0.858 | 143.4 | 17 |
| | PackQViT | 8 | 86.8 | 0.233 | 38.24 | 4.6 |
| | PackQViT | 4 | 44.1 | 0.156 | 26.12 | 3.16 |
| Swin-T | Full-precision | 32 | 114.2 | 0.276 | 44.9 | 5.5 |
| | PackQViT | 8 | 28.6 | 0.099 | 16.4 | 2.1 |
| | PackQViT | 4 | 14.6 | 0.085 | 13.76 | 1.69 |
| Swin-S | Full-precision | 32 | 199.8 | 0.55 | 89.78 | 10.9 |
| | PackQViT | 8 | 50.2 | 0.213 | 34.4 | 4.2 |
| | PackQViT | 4 | 25.5 | 0.164 | 25.93 | 3.29 |

computation). Hence, ViT can obtain more speedup gain than CNN, whose memory movement, such as weight reshaping, image2column, and column2image under different data layouts in each convolution operation, can dilute the efficient quantization speedup on GeMM. The 4-bit compression and concatenation technique can further improve this advantage, achieving approximately 1.75x acceleration compared to INT8 multiplication. This is because, while the theoretical computational workload is halved, overhead is introduced due to internal shifts of concatenated weights and the recovery of stored results in INT8 format.

Our 4-bit packed multiplier can be executed on mainstream processors on edge platforms (e.g., mobile phones, Raspberry Pis, and RISC-V IoT processors), which face challenges in processing low-bit data since their SIMD instructions only support 8-bit or wider data granularity. Hence, we also test the speedup gain on other edge devices. Note that INT8 matrix multiplication within PackQViT utilizes the original byte-level quantized GeMM kernel. For the 8-bit configuration, models can be speedup by up to 3.7x and up to 2.7x for DeiT and Swin models, respectively. For the 4-bit configuration, models can be speedup by up to 5.5x and up to 3.5x for DeiT and Swin models through our proposed packed 4-bit multiplier.

### 5.3 Ablation Study

**Breakdown of Task Accuracy.** We give quantitative results of the proposed log2 quantization, outlier-aware training, and integer activation functions in Table 3. We set Q-ViT as the quantization baseline, which can improve the accuracy by 1.4%, which illustrates the redundancy inside the full precision model and suffers a performance drop by 0.1% in 4-bit settings. Log2 Attention and outlier-aware Training improve the performance when used alone, and integer activation functions almost maintain the original precision with only 0.1% accuracy drops when setting $n=4$ in Int-$2^n$-Softmax. When combining all the components to enable integer-only flow, the performance can be boosted considerably, e.g., log2 quantization improves the 8-bit baseline by 0.5% and outlier-aware training by 0.6%. In contrast, the combination with integer-only functions can reach 0.9%.

## 6 Conclusion and Limitations

This paper proposes an activation-aware fully sub-8-bit QAT framework called PackQViT for ViT inference acceleration on the mobile. We first arrived at several critical observations regarding the data distribution in ViTs and pointed out two distinct characteristics in activations influencing the quantization strategy: a *long-tailed distribution*, and *systematic channel-wise outliers*. To address the long-tailed distribution, we utilized 4-bit log2 quantization and clipping. For the systematic channel-wise outlier, we designed outlier-aware training to predict outliers' indexes and scale and regularize them with PTR in advance. We also develop a SIMD-based 4-bit packed multiplier to support PackQViT and achieve end-to-end ViT acceleration on mobile phones. Experiments show that PackQViT achieves superior task accuracy compared to SOTA quantization studies with significant hardware efficiency. Notably, we primarily validate our QAT methods with 4-bit precision. However, the PackQViT also holds the potential for application in lower precision scenarios. We are currently developing an implementation of a lower-bit SIMD-based multiplier to achieve enhanced acceleration.

## 7 Acknowledgment

This work is supported in part by the National Science Foundation CCF-1937500.

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
