# 8 Appendix

## 8.1 Integer-only Inference

### 8.1.1 Int-LayerNorm

**Inference stage.** During inference, with the gathered $i$ and $r$ of systematic outliers, we extract the layer-wise parameters $s$ and $z$, and compute $\mu$, $\sigma$ in the integer domain. Meanwhile, thanks to the BitShift, PTR $r$ can be calculated with the quantized Layernorm efficiently.

$$\widehat{X_Q} = (X_Q - z) << r, \quad \mu(X) \approx \mu(2^r s \cdot (X_Q - z)) = s \cdot \mu(\widehat{X_Q}), \quad \sigma(X) \approx \sigma(2^r s \cdot (X_Q - z)) = s \cdot \sigma(\widehat{X_Q}). \quad (6)$$

**Integer-only calculations on the statistics.** LayerNorm can be formulated as:

$$LayerNorm(X) = \frac{X - \mu_x}{\sqrt{\sigma_x^2 + \varepsilon}} \cdot \gamma + \beta = \frac{X}{\sqrt{\sigma_x^2 + \varepsilon}} \cdot \gamma + \frac{\gamma\sqrt{\sigma_x^2 + \varepsilon} - \gamma\mu_x}{\sqrt{\sigma_x^2 + \varepsilon}} \quad (7)$$

where $\gamma$, $\beta$ are the learned parameters, and $\mu_x$, and $\sigma_x^2$ are the statistics, which are calculated adaptively based on the input.

Following the definition of LayerNorm, the whole process can be divided into two phases. In the first phase, we shift the quantized activation $X_Q$ with PTR $r$:

$$\widehat{X_Q} = (X_Q - z) << r, \quad (8)$$

In the second phase, we deduce the mean of $X$ and $X^2$ as follows:

$$\mu_x \approx \frac{1}{C}\sum_{i=1}^{C}(\hat{X}_{Q_i} \cdot s) \to \frac{s}{C}T_1, \quad \mu_{x^2} \approx \frac{1}{C}\sum_{i=1}^{C}(\hat{X}_{Q_i} \cdot s)^2 \to \frac{s^2}{C}T_2, \quad (9)$$

Then we deduce the $\sigma_x^2$:

$$\sigma_x^2 = \mu_{x^2} - \mu_x^2 \approx \frac{s^2}{C^2}(CT_2 - T_1^2), \quad \sqrt{\sigma_x^2 + \varepsilon} \approx \frac{s}{C}\sqrt{CT_2 - T_1^2}, \quad (10)$$

Hence, we deploy integer-only calculations to obtain the statistics of input $X$.

**Integer-only Inference.** We rewrite the LayerNorm flows with the quantized $X$ and output $O_Q$:

$$O_Q = \lfloor \frac{s_x\gamma}{\sqrt{\sigma_x^2 + \varepsilon}}\hat{X}_Q + \beta\frac{\sqrt{\sigma_x^2 + \varepsilon} - \gamma\mu}{s_o\sqrt{\sigma_x^2 + \varepsilon}} \rceil + z_o, \quad A = \frac{s_x\gamma}{\sqrt{\sigma_x^2 + \varepsilon}}, \quad B = \beta\frac{\sqrt{\sigma_x^2 + \varepsilon} - \gamma\mu}{s_o\sqrt{\sigma_x^2 + \varepsilon}}, \quad (11)$$

Then for the implementation inference ($b$ is bit-width.):

$$M_1 = b - 1 - \lfloor log_2|A| \rfloor, \quad M_2 = \lfloor |A|2^{M_1} \rfloor, \quad A = sign(A) \cdot \frac{M_2}{2^{M_1}}, \quad O_Q = \lfloor A\hat{X}_Q + B \rfloor + z_o \quad (12)$$

### 8.1.2 Int-$2^n$-Softmax Inference

**Experimental Results of $2^n$ replacement.** Our experiments are conducted on ImageNet-1K [13] with different backbones including DeiT-T, DeiT-S, DeiT-B [43]; Swin-T, Swin-S [33]. The image resolution is $224 \times 224 \times 3$. We follow most of the training settings as in DeiT and train all models for 300 epochs on 4 NVIDIA V100 GPUs. As shown in Table 5, there is no accuracy loss or negligible degradation (<0.01%).

**Integer-only Inference.** We modify the original SoftMax Function in following ways:

$$SoftMax(X) = \frac{exp(X_i)}{\sum_{j=1}^{J}exp(X_j)} \to \frac{2^{X_i}}{\sum_{j=1}^{J}2^{X_j}}, \quad (13)$$

Then we subtract the maximum input to avoid overflowing and rewrite the $2^n$-SoftMax:

$$\hat{X}_i = X_i - max(X_i), \quad 2^n\text{-SoftMax}(X) = \frac{2^{\hat{X}_i}}{\sum_{j=1}^{J}2^{\hat{X}_j}}, \quad (14)$$

Since the input $X$ is the integer value, combined with the BitShit of $2^n$ to replace the multiplication of the exponential function, we execute the integer-only computation.

Table 5: The training results of $2^n$ replacement in the Int-SoftMax on the ImageNet-1K.

| Method | DeiT-T | DeiT-T-$2^n$ | DeiT-S | DeiT-S-$2^n$ |
|--------|--------|-----------|--------|-----------|
| **Accuracy** | 72.21 | 72.23 | 79.85 | 79.85 |
| **Method** | DeiT-B | DeiT-B-$2^n$ | Swin-T | Swin-T-$2^n$ |
| **Accuracy** | 81.85 | 81.86 | 81.35 | 81.34 |
| **Method** | Swin-S | Swin-S-$2^n$ | | |
| **Accuracy** | 83.2 | 83.18 | | |

Table 6: Training cost for PackQViT .

| Model | #Head | Embed. Dim | Depth | Baseline hours | Training hours | Training Saving |
|-------|-------|-----------|-------|---------------|---------------|-----------------|
| DeiT-T | 3 | 192 | 12 | 24.8 | 17.2 | 35.60% |
| DeiT-S | 6 | 384 | 12 | 44.6 | 26.8 | 39.90% |
| DeiT-B | 12 | 768 | 16 | 168.2 | 79.1 | 53.20% |
| Swin-T | [3,6, 12,24] | [96, 192, 384, 768] | [2, 2, 6, 2] | 35.6 | 27.4 | 26.70% |
| Swin-S | [3,6, 12,24] | [96, 192, 384, 768] | [2, 2, 18, 2] | 114.5 | 62.4 | 45.40% |

### 8.1.3 Training Comparison between Quantization Fine-tuning and Baseline Train-From-Strach

Because we execute the outlier-aware training in a fine-tuned way, i.e., 70 epochs for step 2, 30 epochs for step 1, the training hours of the entire pipeline can be 26.7%~54% of train-from-scratch of full precision models.

### 8.1.4 Systematic Channel-Wise Outliers of ViTs

We present the systematic channel-wise minimum and maximum values of ViTs as shown in Figure 7. For comparison, we choose the input of the last LayerNorm layer for ViTs. We test 1024 images, and outliers always occurred in fixed indexes. It is observed that serious and systematic inter-channel variations are found in ViTs.

### 8.1.5 CPU Profiling of Popular ViT Models in FP32, INT8 and INT4 Precision

In this section, we show more details in the hardware profiling of ViT models, which mainly includes FP32, INT8 w/o quantization, full INT8, INT4 w/o quantization, and full INT4, as shown in Figure 8 - 12 (platform: Snapdragon 870 SoC CPU).

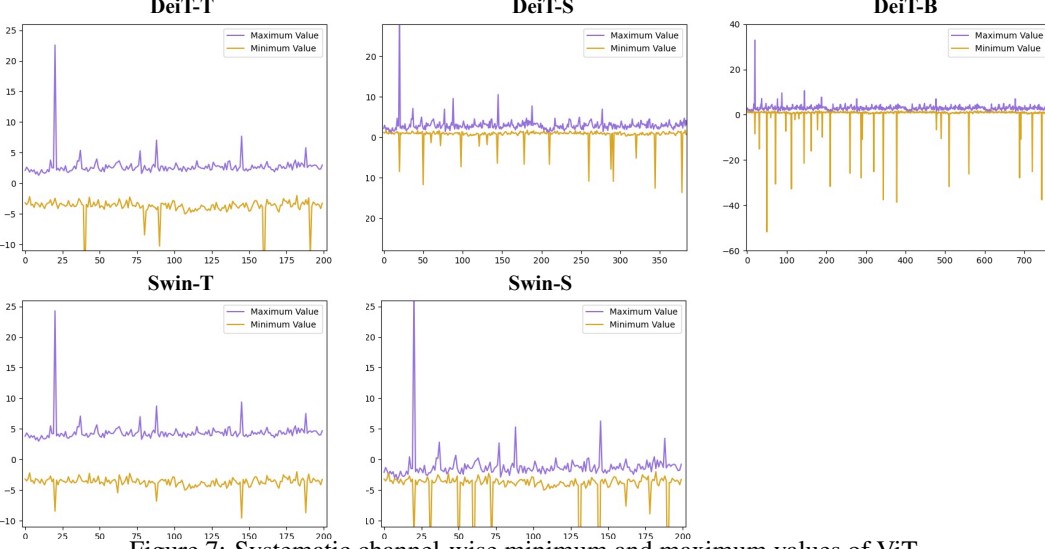

Figure 7: Systematic channel-wise minimum and maximum values of ViT.

Table 7: Operator speedup compared with the original implementation.

| Operator | Snapdragon 870 onboard CPU |
|----------|---------------------------|
| Softmax | 3.7x |
| GeLU | 3.9x |
| Layernorm | 1.4x |

Table 8: Evaluating non-linear operator optimization gain on DeiT-T.

| Method | Latency(ms) |
|--------|-------------|
| Int4-w/o FullQuantization | 46.7 |
| + Int-24-Softmax | 41.2 |
| + I-GeLU | 36.6 |
| + Int-Layernorm | 34.8 |

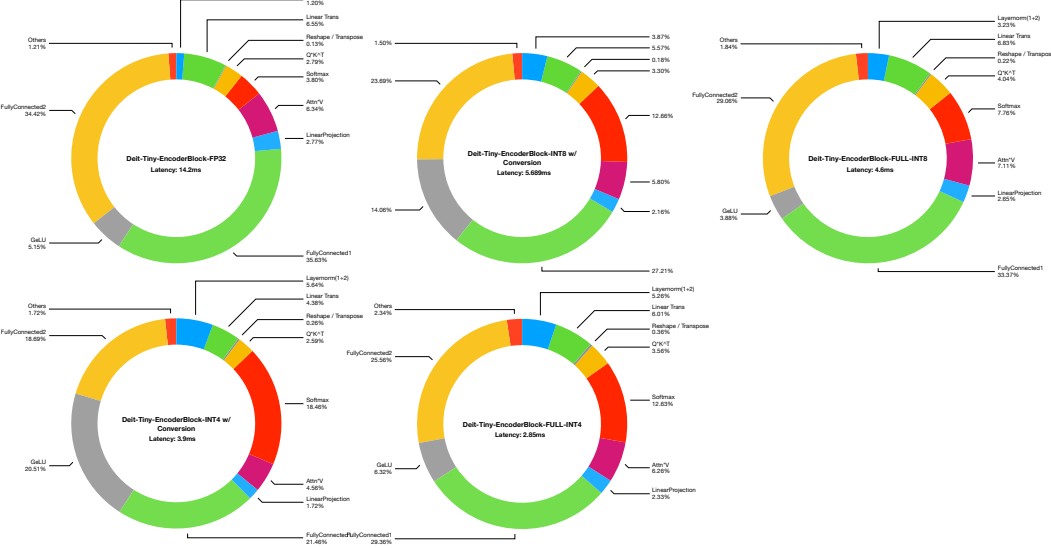

Figure 8: CPU profiling of DeiT-Tiny.

### 8.1.6 Breakdown of Efficiency Gain

Table 7 shows the actual speedup on Arm-based platforms of our hardware-friendly design of nonlinear operations. As shown in Table 8, the integer-based optimization on 3 non-linear operators would provide an overall latency decrease of approximately 12ms, more than 25% of the original non-optimal method. Eliminating non-linear operators built on floating-point instructions from the overall computation flow has significantly improved efficiency. This optimization becomes even more meaningful when applied alongside 4-bit quantization, where non-linear operations share a higher latency proportion with further acceleration on matrix multiplication.

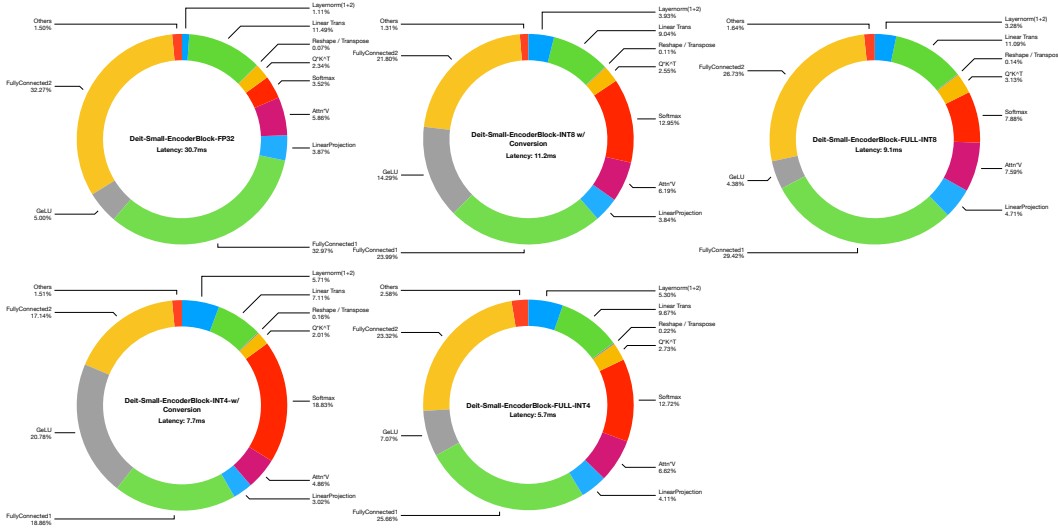

Figure 9: CPU profiling of DeiT-Small.

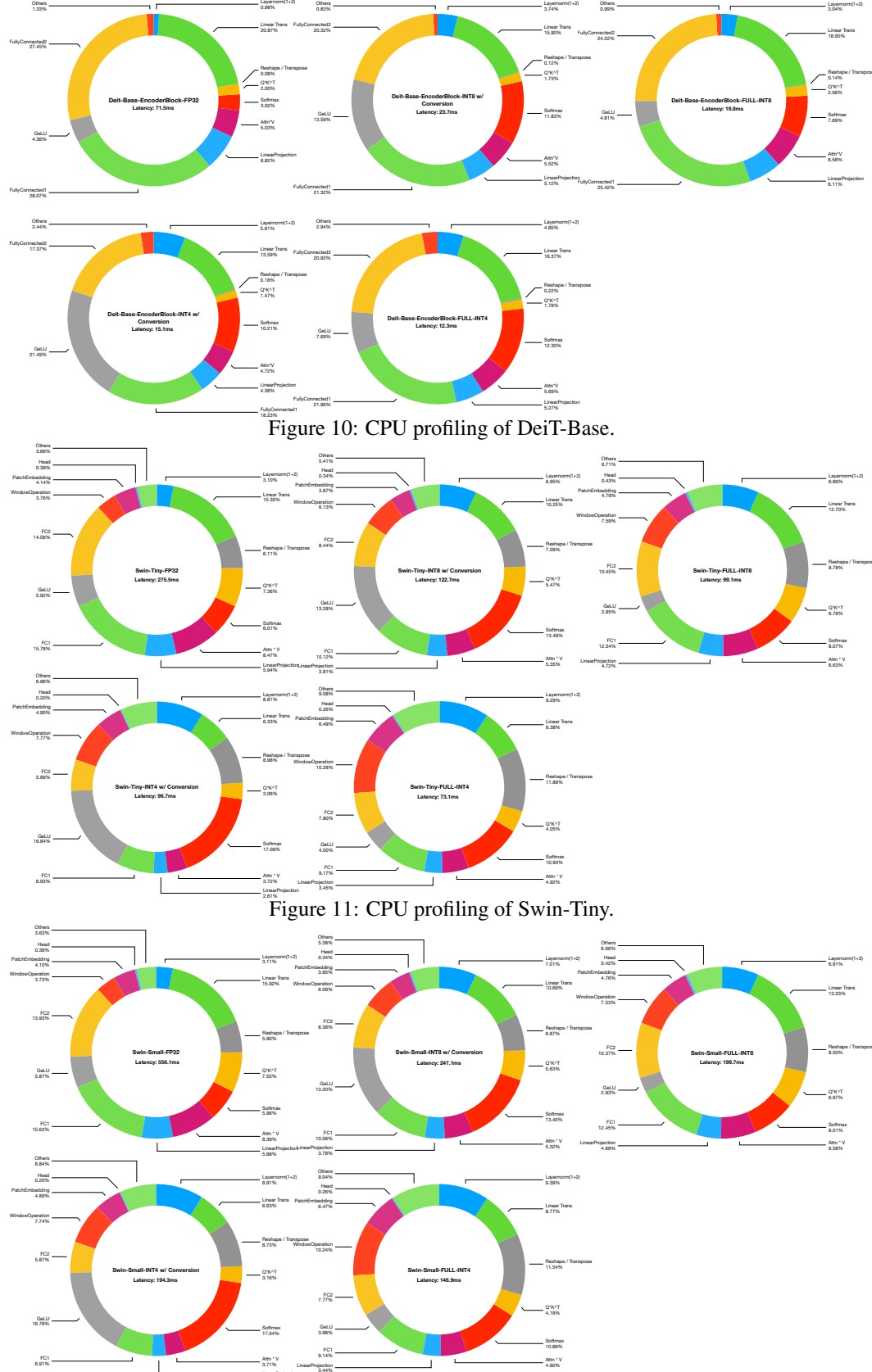

Figure 10: CPU profiling of DeiT-Base.

Figure 11: CPU profiling of Swin-Tiny.

Figure 12: CPU profiling of Swin-Small.