# OpenReview forum: "PackQViT: Faster Sub-8-bit Vision Transformers via Full and Packed Quantization on the Mobile"
_NeurIPS.cc/2023/Conference — NeurIPS 2023 poster_

### Official Review · Reviewer_ysuz · 2023-07-04

**Soundness:** 2 fair
**Presentation:** 3 good
**Contribution:** 2 fair
**Rating:** 4
**Confidence:** 4

**Summary:**

The paper proposes a 4-bit vision Transformer and claims its efficient deployment on mobile CPUs. The attention module is quantized with mixed quantization schemes, i.e., inputs are quantized using different methods, logarithmic and uniform linear. To run 4-bit inference on CPUs, the paper also presents a manually designed packed matmul kernel that leverages the SIMD instructions.

**Strengths:**

This work managed to quantize a vision Transformer to 4-bit. Most importantly, it runs efficient inference on CPUs, which only has supports for up to 8-bit instructions. The demonstration of the potential of the 4-bit integer quantization for mobile platforms is nice to the community. The latency breakdown of a vision Transformer provides good insights.

**Weaknesses:**

My main concern is the lack of details on the quantization methodology. I will list them as follows:

1. The paper claims integer-only computation flow, but does not explain how this is achieved. For example, how exactly is the GELU function quantized? To achieve a good model quality, quantization functions usually introduce a scaling factor for pre-quantization and post-quantization steps, which includes floating-point operations. Are there such scaling operations in GELU and shortcut quantization here? What is the overhead then? In Figure 4 (right), why are outputs INT8? The outputs correspond to Equation 3 and the accumulation should be high-precision. Are they quantized again? These details are missing.

2. In Section 4.2, the BitShift operation is introduced to compute the matmul between attention scores and value matrices. This is a new operation other than matmul. It is unclear how this is accelerated on CPUs. If another kernel is designed specifically for this operation, how does its performance compare to INT8 SIMD matmul?

3. It is unclear how much benefit quantizing shortcuts can bring since its overhead may be overwhelming for elementwise operations. There needs to be some experimental analysis on this. If the benefits are more on the memory side, then the paper needs to explain the memory layout of the shortcuts, e.g., are the integer shortcuts packed on the fly when transferring to the DRAM?

**Questions:**

The main concerns have already included several questions. More questions that can improve the paper quality if addressed are listed as follows:

1. Line 176 claims that Softmax and GELU lead to the disparity of channel-wise outliers between ViTs and CNNs. This is questionable because (1) softmax is bounded, so it would not cause outliers as shown in Figure 4 (left); (2) GeLU is very similar to ReLU in terms of forward propagation. It does not explain the disparity.

2. Line 203 claims that "GELU truncates activation values to be centered around 0". This is questionable because again GELU is very similar to ReLU. It has no centering effect.

3. What is the model size in the experiments?

4. In Section 5.3 ablation study, is there any explanation on why combing all int quantization will have quality improvement compared to used alone?

**Limitations:**

The paper discusses limitations in Section 6. The reviewer is not aware of any social impact of this paper.

---

> ### Author Rebuttal · Authors · 2023-08-10
>
> **ReW1.**: (1) For efficient implementation of GeLU, we revisit I-GeLU [5] segmentation function with a=-0.2888, b=-1.769, x$\in$Z as Eq. 1 in our appended PDF.
>
> Within the (-2,2) data range, the value of  I-GeLU is {-0.0257, -0.1152, 0, 0.5919, 1.3885} under the integer domain.
> So we directly use their quantized value, which is equal to adapting Eq. 1 into a segmentation function with linear kernels and lookup table under INT scenarios as Eq. 2. Before Int-GeLU$_{imp}$, we do the quantization and just send the integer value directly into these modified nonlinear functions. We can also minimize the quantization error of this modification through our QAT training.
>
> Implementation flow for I-GeLU$_{imp}$ kernel based on ArmISAs:
>
> 	I_GeLU_imp(in_ptr, length):
>     threshold = 3
>     oneVec  = 1
>     zeroVec = 0
>     for i = 0 to length, step 16:
>         /* 16 operands  per loop */
>         vec = load(in_ptr + i)
>         /*  set as 1 when  input = 2 */
>         mask1 = check_equal(vec, 2)
>         vec = select(mask1, oneVec, vec)
>         /* Mask-based BitwiseZeroSetting below threshold */
>         mask2 = compare(vec, threshold)
>         vec = select(mask2, zeroVec, vec)
>         /* Result Storage */
>         store(in_ptr + i, vec)
> , where the actual execution only utilizes efficient integer-based comparison and bitwise selection intrinsics like vceqq_s8 and  vbslq_s8 on ArmISAs instead of complicated floating-point-level raw GeLU calculations.
>
> (2)For the proceeding on the shortcut/residual link: The integer-only means we conduct the inner-operator computation based on the integer computation flow. We currently cannot avoid the floating-point based inter-operator data transmission due to the unavoidable ‘Dequantize’ process of common inference frameworks that would require a predefined scale & zeropoint for ‘integer to floating’ conversion after each operator execution. Also, the latency occupied by residual link addition is trivial, and making it into an integer domain won’t lead to obvious high hardware benefits. In our paper, we do the quantization on the addition results of the residual link, but the addition itself is still computed in FP32. We will revise the ‘integer-only data flow’ into ‘integer-only inner operation inside each computing kernel, which makes the main computation flow keep integer computing.”
>
> (3) For Figure 4, you are right and that value should be INT12. However, we clip the overfitting number into the maximum of INT8, 127.
>
> **ReW2.**: Benchmark on Multiplication Versus BitShifting: We have benchmarked the instruction-level of integer-based multiplication versus BitShifting on on-board Snapdragon 870 SoC.
> Firstly, naive instructions have been deployed: where a * b (b = 4) versus a << 2, each operation has been conducted for 100k iterations for stable latency measurement, based on 10 separate tests we’ve done, Bitshifting generally achieves 1.71x speedup over multiplication. Then, with NEON SIMD support on 128-bit processing registers where ‘vmulq_s8’ (vector_multiplication) and ‘vshlq_n_s8’ (vector_left_shift_lanewise) have been utilized, based on 10 separate tests we’ve done, Bitshifting generally achieves 1.62x speedup over multiplication. Thus, Matmul based on Bitshift operation is friendly to CPUs instead of calling another GeMM kernel, which aligns with the idea of computation efficiency of our article.
>
> **ReW3.**: Just as we mentioned in ReW1, we keep residual links in the floating-point domain and just quantize the addition value of residual links into the integer domain. Sorry for the confusing points again!
>
> **ReQ1.**: In Line 177, we mentioned, “This is because the data flow of nonlinear functions, such as Softmax and GELU, existing in the residual link of transformer structures, results in a greater disparity between the input and output of the residual blocks compared to CNN models. ” We should change ‘such as Softmax and GELU’ into “i.e., LayerNorm, Softmax and GELU.’ According to our visualization (Figure B), these three operators can generate different dynamic ranges of their input/output activation tensors, especially for LayerNorm and GELU. The cumulative difference will result in a considerable mismatch between the different dynamic ranges of activation tensors in the residual connections. And this potentially excites significant outliers in the data flow [9]. We will clarify it in our final version.
>
> **ReQ2.**: We want to mention the long-tail distribution caused by GeLU as Eq. 1, which makes much value turn into 0, as shown in Figure C. We will make this clearer in our final version. Thank you for such a thorough review. You help us a lot to improve our paper presentation.
>
> **ReQ3.**: Please refer to Reviewer QVMx: ReW1.
>
> **ReQ4.**: Our baseline is another QAT method, Q-ViT. Each ‘+’ in Table 3 of Section 5.3 means we correct one disadvantage of Q-ViT. After each added technique, the accuracy compared with baseline, Q-ViT, can be improved. So after combining them all together, the improved accuracy can be better than one trick used alone. Thanks again for your careful and responsible review.
>
>  Your comments really help us improve our paper and also give us deeper thinking on ViT structures and on-device implementation. Because of the limited pages, we are sorry that we cannot elaborate on the details.  We will add more in the appendix in our final version.
>
> [9] Bondarenko,  et al. "Understanding and overcoming the challenges of efficient transformer quantization." arXiv preprint arXiv:2109.12948 (2021).

---

> > ### Author Response · Authors · 2023-08-17
> > **Further Discussions with Reviewer ysuz**
> >
> > Dear Reviewer ysuz:
> >
> > Thank you once again for dedicating your time and effort to evaluating our paper. We hope that our rebuttal addresses the concerns you have. If you have any remaining questions or reservations, we are more than willing to engage in further discussion. Your contribution to enhancing the quality of our paper is greatly appreciated.

---

> > ### Comment · Area_Chair_brkv · 2023-08-18
> >
> > Dear reviewer,
> >
> > The discussion with authors is closing soon, please review the rebuttal to see if the authors have addressed your concerns, and acknowledge to the authors that you have read their response.
> >
> > Thanks

---

> > ### Comment · Reviewer_ysuz · 2023-08-19
> > **Response to rebuttal**
> >
> > Thanks for your answering.
> >
> > However, the concern about the BitShift matmul kernel is not addressed, which seems to be a key to the overall speedup.
> >
> > In the rebuttal, the authors benchmarked a single shift instruction on CPU and compared it to a (INT8?) multiplication. This is actually not that informative. Note that on CPUs matmul is not a simple multiplication within for loops. Otherwise, the control logic overhead will be much higher than the computation itself. CPUs usually have instructions that can do multiple multiply-adds within one cycle, e.g., VPMADDUBSW on x86 CPUs. There matmuls start to get a decent speedup. The real question here is that the authors did not present a clear design of the log2-integer matmul kernel, but claimed a speedup compared to the INT8 intrinsic CPU matmul, which is not convincing.

---

> > > ### Author Response · Authors · 2023-08-19
> > > **Further Discussions with Reviewer ysuz**
> > >
> > > **Re**: Thanks for your further questions to afford us to supply more details on our hardware implementation. We totally agree with you that ‘CPUs usually have instructions that can do multiple multiply-adds within one.’
> > > In ARM ISAs, the similar instruction is ‘vmla’ within the NEON SIMD (simple-instruction-multiple-data) intrinsics collection, which means ‘vector_multiply_add’. Meanwhile, according to our test based on ArmComputeLibrary cross-compiled for Android-armv8 with AndroidNDK 20.1, these two methods, ‘multiply-adds together’ and ‘multiply-add separated’ exhibit similar latency with the same matmul workload, as shown in Table I, where ‘multiply-add separated’ utilizes vadd (c, vmul(a, b)) instead of vmla(c, a, b) inside GeMM kernel while keeping other parts the exact same, e.g.,  memory reshaping with Interleave & Transpose before mathematic computations.
> > >
> > > Table I. Latency on INT8 Square MM
> > > |Methods | Latency on SquareMatmul of (1000,1000) (ms)|
> > > |------------|---------------|
> > > |VMLA	|12.8	|
> > > |VMUL+VADD	|13.0	|
> > >
> > > Furthermore, NEON SIMD support also covers bitshifting operation in ARM ISAs, ‘vshl’ left shifts each element in a vector by an amount specified in the corresponding element.
> > > Thus, our ‘BitShift matmul kernel’ is not a direct design with nested loops, but a refined-modification on the existing GeMM kernel structure, where the multiplication (vmul) has been substituted with BitShifting (vshl) for upgraded efficiency. This means our instruction-level benchmark results are informative and effective as well.
> > >
> > > We also provide the ‘BitShift matmul kernel’ as follows, where the inner_window_loop for 4x4 block computation has been displayed.
> > >
> > > log2_bitshift_GeMM_4x4(i, s1, s2):
> > >
> > >     s1 = [ 4x4 matrix of src1 ]
> > >
> > >     s2 = [ 1x4 vector of src2, all value ]
> > >
> > >     c = [ 4x4 matrix of zeros ]
> > >
> > >     p = [ 1x4 vector with zeros ]
> > >
> > >     /* Inner loop for 16 elements 4 units per loop */
> > >
> > >     for j in range(4):
> > >
> > >         /* Lane-wise BitShift */
> > >
> > >         p = [LeftShift the j-th row of s1 with the i-th element of s2]
> > >
> > >         /* Accumulation */
> > >
> > >         c.row[j] = [add elements of p to the corresponding j-th row of c]
> > > return c
> > >
> > >
> > > In conjunction with thorough on-device testing, we make direct modifications and optimizations to the computing kernel, utilizing the existing instruction set and GeMM engine. Additionally, when incorporating system-level optimizations, the outcomes could potentially vary, yet implying a substantial workload overhead.
> > >
> > > We appreciate your enthusiastic engagement and insightful inquiries. Your rigorous questioning has significantly contributed to the enhancement of our work.

---

### Official Review · Reviewer_R5sB · 2023-07-06

**Soundness:** 3 good
**Presentation:** 3 good
**Contribution:** 3 good
**Rating:** 6
**Confidence:** 4

**Summary:**

This paper is about developing a quantization method for ViT, including DeiT and Swin Transformer. The developed algorithm can be run on mobile phones. Experiments were conducted on several datasets, with comparisons to other quantization methods.

**Strengths:**

a new strategy for quantization of ViT, mainly for inference;

saving computation cost and memory usage in inference.

**Weaknesses:**

The presentation is not clear in some places. The authors should re-organize the parts of introduction and method, making it clearer and easy understanding for readers.

**Questions:**

no other questions. See the weakness part for comments.

**Limitations:**

it may reduce the accuracy for some application scenarios, using quantization.

---

> ### Author Rebuttal · Authors · 2023-08-09
>
> **ReW1**: Thank you so much for improving our paper presentation. We will rewrite the final paragraph of our introduction as follows:
>
> Our contribution can be categorized into four aspects, including motivations, optimization of task accuracy, optimization of hardware efficiency, and evaluation results.
>
> Motivation: We first perform the analysis of data distribution within ViTs (long-tail distribution and outlier analysis), which provides insights on quantizing model weights and activations effectively in order to enhance task accuracy and on-device efficiency. Meanwhile, we conduct a hardware latency profiling to figure out the runtime bottleneck of the whole models, which guides us to optimize operator kernels for efficiency.
>
> Optimization of task accuracy: We apply the log2 on attention to deal with the long-tail distribution of the attention matrix and deploy the outlier-aware training to eliminate accuracy degradation caused by outlier quantization.
>
> Optimization of hardware efficiency: (1) We figure out the nonlinear operators as the speed bottleneck. Then, we adapt them into hardware-friendly ones. Int-2n-Softmax, Int-GeLU, Int-layerNorm; (2) We leverage structured/fixed patterns of data outliers to predict their channel index and power-of-two ratio in advance through training. Then we can preload these indexes and ratios with the model weight and compute them at the same time with the matrix multiplication between weights and activations. Thus, our design can avoid dynamic hardware overhead; (3) We quantize models into 4-bit and propose the packed 4-bit multiplier to support this sub-8-bit matrix multiplication.
>
> Evaluation results: We not only can get superior accuracy results than existing works but also accelerate end-to-end on-device latency performance of the quantized ViTs on multiple edge devices.
>
> Meanwhile, following these organizational logic motivations, optimization of task accuracy, and optimization of hardware efficiency, we will restructure the presentation of our methodology.
>
> Thank you for supplying this chance for us to make our paper clearer.
>
> **ReLimitation**: Thanks for your kind reminder. Please refer to Reviewer yEDd: ReW3 for the detailed results on object detection. Indeed, it shows that for more complex tasks, quantization tends to lead to a degradation of task performance.
> Thanks again for your valuable questions. In our final version, we will add one discussion part to clarify this point.

---

> > ### Comment · Area_Chair_brkv · 2023-08-18
> >
> > Dear reviewer,
> >
> > The discussion with authors is closing soon, please review the rebuttal to see if the authors have addressed your concerns, and acknowledge to the authors that you have read their response.
> >
> > Thanks

---

### Official Review · Reviewer_ktdW · 2023-07-06

**Soundness:** 2 fair
**Presentation:** 2 fair
**Contribution:** 2 fair
**Rating:** 5
**Confidence:** 4

**Summary:**

This paper introduces a quantisation framework for end-to-end ViT inference on CPU devices. The main idea is to exploit the byte-level data parallelism present in the SIMD units of CPUs, and present a comprehensive set of techniques to enable low-latency ViT inference. The proposed framework includes two parts: quantization tricks , and a data-parallel strategy using SIMD instructions. To prove the efficacy of the proposed framework, the paper presents an accuracy evaluation (Table 1) and a performance for the ViT inference on CPUs (Table 2).

**Strengths:**

The paper has successfully demonstrated improved latency on mobile devices, verifying that these results are not just theoretical. Furthermore, the paper highlights the great performance achieved in the QAT setup (see Table 1). The major benefit of this approach is the reduced engineering effort when packing items into SIMD units.

**Weaknesses:**

The two major issues with this paper are firstly, its evaluation, and secondly, its originality. Regarding its evaluation, the paper is a QAT method but it is only compared to various PTQ methods in Table1. This is not a fair comparison unless you showcase the PTQ performance under the same precision setup. Furthermore, I am uncertain how much accuracy each method is providing (e.g. int-layernorm, log2 quan...).

This paper lacks novelty, as the proposed quantisation is merely an amalgamation of existing methods. Specifically, treating channel-wise outliers is widely discussed in LLM quantisation work (e.g. LLM.int8()) and various quantisation methods in the CNN domain. Furthermore, the int-layernorm is also from existing work. The only notably new contribution appears to be log2 quantisation on activations and softmax in my opinion, although this is not focused on in the paper, limiting its novelty claim.

The presentation is far from its best state. The color schemes of this paper are suboptimal (e.g., Figures 1 and 2). Generally, all fonts used in the figures are too small and this really impairs readability. Furthermore, names such as "Int8 No full" in Figure 1 are quite confusing. Additionally, Figure 4 is too small and the rightmost figure is overlapped with the one in the middle.

**Questions:**

- How do you see the proposed approach being used as PTQ? If it cannot be used as a PTQ method, why? If it can, how does it compare to other methods in Table 1?
- Why would this only work on CPUs? What about the INT8 support on GPUs?
- What performance gain does each optimisation provide? For example, can you isolate each one and explain how much performance gain it delivers?

**Limitations:**

I do not think this is applicable to this paper.

---

> ### Author Rebuttal · Authors · 2023-08-09
>
> **ReW1**: Thanks for your questions helping us clarify the paper. (1) We have reported our results with QAT in the lower part of Table 1 in our paper. We also provide our analysis of each method in the ablation study, as shown in Table 3 of our paper. (2) Our approach is compatible with PTQ. We report our method only under the PTQ setting as shown in Table E. Since the accuracy optimization of outlier-aware training cannot be leveraged, our results are comparable to the SOTA PTQ methods.
>
> **Table E.** Comparison of the Top-1 (%) accuracy with the SOTA PTQ method on the ImageNet dataset.
> | Method | #Bits | DeiT-T | DeiT-S | Swin-T | Swin-S |
> |-----------|-------------------|-----|-----|-----|------|
> | FQ-VIT | 8/8/8 | 71.6 | 77.1 | 81.2 | 80.5 | 82.7 |
> | ours      | 8/8/8 | 71.4 | 77.4 | 81.4 | 80.1 | 82.7 |
>
> **ReW2**: (1) LLM.int8() is a literature on a large language model, Bert. Thanks to its inspiration, we conducted a series of further explorations on ViT models.
> LLM.int8() utilizes per-channel quantization with mixed precision to reduce the adverse outlier effect of activations. However, per-channel quantization is not a hardware-friendly way and is not supported in current main computing frameworks, as mentioned in Reviewer 7eJH: Q2. In contrast, we deploy outlier-aware training and layer-wise quantization with power-of-two coefficients on the outlier channel, which enhances the task accuracy with negligible hardware overhead.
>
> (2) Please refer to Reviewer 7eJH: ReW1 which we clarify our novelties.
>
> **ReW3**: “INT8 No full” means we only quantize weights and activations in matrix multiplication, while leaving the nonlinear operations, e.g., GeLU, Softmax, and LayerNorm still in floating-point domain computation. We will change “int8 No full” in Figure 1 into “INT8 w/o FullQ” with detailed notes. We draw Figure 2 again to make our hardware profiling clearer.  Also, we make Figures 1 and 4 clearer and upload these figures in our appended PDF. We apologize for our ambiguity. And we will make it clearer in our final paper version.
>
> **ReQ1**: Please refer to W1.
>
> **ReQ2**: Currently, integer-based SIMD support is widely adopted at the CPU level. Many mainstream inference engines utilize integer-based SIMD acceleration for quantized linear operators, leading to significant reductions in latency. However, current mobile GPUs offer varying levels of support for INT8 instructions. For example, Arm has started supporting 8-bit sdot from Mali G71, G52, etc., while still remaining unsupported for other series. Thus, current quantized inference on GPU still lacks a standardized design framework. To our knowledge, no inference engine utilizes genuine 8-bit integer instructions to construct the GPU inference kernel. SUMMIN KIM and his team pointed out that, ‘there are currently few frameworks that support INT8 quantization for mobile GPUs’ in “Performance Evaluation of INT8 Quantized Inference on Mobile GPUs,” published on IEEE Access. As claimed in the official doc of TensorflowLite, they claimed that ‘we run quantized models by giving it a ‘floating-point view’ of the original model’ based on floating-point execution backend. Besides, the supporting document of ArmComputeLibrary illustrates the requirement of INT8 conversion to INT32 before actual execution. Thus, our computation flow has not yet been promoted onto quantized inference on mobile GPUs.
>
> The framework we propose is a versatile solution that, in theory, can be adapted to various backends that support true integer-based fixed-point inference. However, we have not yet extended its application to the mobile GPU domain due to the lack of sufficient hardware integer-processing capability or slow integration of computation libraries on mobile GPUs.
>
> But that shall be the next stage of our research. We will first try to establish an applicable and generalizable quantization framework for mobile GPU, including both a new quantization algorithm design and int8-oriented kernel implementation. Then, we would expand the low-bit inference onto the mobile GPU field.
>
> **ReQ3**: Thanks for helping us to break down our optimization gain in detail both from the accuracy and latency part.
>
> (1) For the breakdown analysis of our accuracy, please refer to the ablation study and Table 3 in our paper, where we break down our accuracy gain for each sub-method.
> (2) For the optimization of hardware efficiency, you can refer to our latency profiling Figure in Q3. Also, Table F shows the actual speedup on Arm-based platforms of our hardware-friendly design of nonlinear operations.
>
> Table F. Operator speedUp compared with the original implementation.
> | Operator                      | Snapdragon 870 onboard CPU |
> |------------------------------ | ------------------------- |
> | Softmax                       | 3.7x                      |
> | GeLU                          | 3.9x                      |
> | Layernorm                     | 1.4x                      |
>
>  Table G. Evaluating non-linear operator optimization gain on DeiT-T.
> | Method                      | Latency(ms) |
> |------------------------------ | ------------------------- |
> | Int4-w/o FullQuantization                       | 46.7                       |
> | +  Int-$2^{4}$-Softmax                          | 41.2                      |
> | + I-GeLU                     | 36.6                      |
> | + Int-Layernorm               | 34.8        |
>
> As shown in Table G, the integer-based optimization on 3 non-linear operators would provide an overall latency decrease of approximately 12ms, more than 25% of the original non-optimal method. The elimination of non-linear operators built on floating-point instructions from the overall computation flow has led to significant efficiency improvements. This optimization becomes even more meaningful when applied alongside 4-bit quantization, where non-linear operations share a higher latency proportion with further acceleration on matrix multiplication.

---

> > ### Comment · Reviewer_ktdW · 2023-08-15
> >
> > Thank you for providing the detailed breakdowns, but some of the replys caused further confusion on my side.
> >
> > > ReW1: Thanks for your questions helping us clarify the paper. (1) We have reported our results with QAT in the lower part of Table 1 in our paper. We also provide our analysis of each method in the ablation study, as shown in Table 3 of our paper. (2) Our approach is compatible with PTQ. We report our method only under the PTQ setting as shown in Table E. Since the accuracy optimization of outlier-aware training cannot be leveraged, our results are comparable to the SOTA PTQ methods.
> >
> > What I am saying is they way you present results in Table 1 is confusing, if you are presenting the QAT mode of your method, adding PTQ methods in the top-half is misleading. If you have the PTQ compatible version, then you should put it on the last row under PTQ. There are also a range of QAT method proposed after the Q-vit paper, the most recent maybe is [1]. I think these methods should be included to give a fair comparison, and also your PTQ results should be added to the top half.
> >
> > > ReQ2: Currently, integer-based SIMD support is widely adopted at the CPU level. Many mainstream inference engines utilize integer-based SIMD acceleration for quantized linear operators, leading to significant reductions in latency. However, current mobile GPUs offer varying levels of support for INT8 instructions. For example, Arm has started supporting 8-bit sdot from Mali G71, G52, etc., while still remaining unsupported for other series. Thus, current quantized inference on GPU still lacks a standardized design framework. To our knowledge, no inference engine utilizes genuine 8-bit integer instructions to construct the GPU inference kernel. SUMMIN KIM and his team pointed out that, ‘there are currently few frameworks that support INT8 quantization for mobile GPUs’ in “Performance Evaluation of INT8 Quantized Inference on Mobile GPUs,” published on IEEE Access. As claimed in the official doc of TensorflowLite, they claimed that ‘we run quantized models by giving it a ‘floating-point view’ of the original model’ based on floating-point execution backend. Besides, the supporting document of ArmComputeLibrary illustrates the requirement of INT8 conversion to INT32 before actual execution. Thus, our computation flow has not yet been promoted onto quantized inference on mobile GPUs.
> >
> > I see your point about the aim of Tensorflow-lite to initially support "emulated quantization". However, I am a bit confused about the argument being made in relation to this. From my understanding, Nvidia's TensorRT is capable of supporting int8 inference without much difficulty. Therefore, I fail to grasp the entire argument about mobile GPUs lacking this type of support and referencing the ARM ecosystem, in fact, [1] has also shown real speed-ups on Nvidia devices.
> >
> > [1] https://arxiv.org/abs/2305.10727
> > [2] https://developer.nvidia.com/blog/achieving-fp32-accuracy-for-int8-inference-using-quantization-aware-training-with-tensorrt/

---

> > > ### Author Response · Authors · 2023-08-17
> > > **Further Discussions with Reviewer ktdW**
> > >
> > > **Re-ReW1**: Thanks for your valuable suggestions, which will greatly improve our paper presentation. As you suggested, we will add all the results in our final version as follows:
> > >
> > > Table H. Comparison of the Top-1 (%) accuracy with state-of-the-art methods on the ImageNet dataset. We report bit width in the order of weight/activation/attention.
> > > | Method | #Bits | DeiT-T | DeiT-S |DeiT-B| Swin-T | Swin-S |
> > > |--------------------|-------------|---------|---------|--------|--------|--------|
> > > | Full Precision | 32/32/32 | 72.21 | 79.85| 81.85| 81.35 | 83.2 |
> > > |		                 PTQ			          |
> > > | MinMax          | 8/8/8    | 70.94 | 75.05 | 78.02| 64.38 | 74.37 |
> > > | EMA               | 8/8/8    | 71.17 | 75.71 | 78.82| 70.81 | 75.05 |
> > > |Percentile        | 8/8/8    | 71.47 | 76.57 | 78.37| 78.78 | 78.12 |
> > > |OMSE             | 8/8/8     | 71.3   | 75.03| 79.57| 79.3   | 78.96  |
> > > |Bit-Split           | 8/8/8     | —   | 77.06| 79.42| —    | —   |
> > > |PTQ for ViT      | 8/8/8     | —   | 77.47| 80.48| —    | —   |
> > > |FP-ViT             | 8/8/8     |71.61 | 77.06| 81.2|80.51   | 82.71   |
> > > |PackQViT        | 8/8/8     |71.44 | 77.43| 81.4|80.18   | 82.74  |
> > > |		                 QAT			          |
> > > |GPUSQ-ViT[1]  | 8/8/8     | 72.4  | 80.3| 82.9| 81.2   | 83.1  |
> > > |Q-ViT                | 8/8/8     | 73.6  | 80.2| 82.4| 81.8   | 83.6  |
> > > |PackQViT         | 8/8/8     | 74.6  | 80.8| 82.9| 82.4   | 84.1  |
> > > |PackQViT         | 8/8/4     | 74.5  | 80.8| 82.9| 82.3   | 84.1  |
> > > |GPUSQ-ViT[1]  | 4/4/4     | 71.7  | 79.3| 81.6| 80.7   | 82.8  |
> > > |Q-ViT                | 4/4/4    | 72.1  | 79.1| 81.1| 81   | 82.4  |
> > > |PackQViT         | 4/4/4     | 72.7  | 79.6| 81.5| 81.5   | 82.8  |
> > >
> > > As shown in Table H, our proposed PackQViT can consistently outperform GPUSQ-ViT by up to 2.2% and 1.2 % for DeiT and Swin, respectively, under the 8-bit precision and can be comparable or better than (up to 1% and 0.8% for DeiT and Swin, respectively) GPUSQ-ViT under the 4-bit precision.
> > >
> > > **Re-ReW2**: Thanks for your following questions, which afford us to clarify it further. Currently, NVIDIA does not offer any mobile GPU products. Hence, NVIDIA TensorRT (Tensor Core) cannot be utilized for inference on mobile GPUs [3]. Specifically, Tensor Core can support the GPU architecture: Hopper (e.g., H100), Ampere (e.g., A100, A6000), Turing (RTX2080, GeForce 20 series), Volta (e.g., V100) [4]. The GPUs used in [1] are A100 GPU and AGX Orin, both of which are high-end NVIDIA GPUs and utilize brand-new Ampere architecture. Hence, [1] can leverage the TensorRT engine. The GPUs used in [1] do not serve mobile devices and also edge devices which are our research target.
> > >
> > > Presently, there are two primary challenges in implementing INT8 inference on mobile GPUs: native architecture support and mainstream inference engine compatibility.
> > >
> > > Taking smartphones as an example, the integrated mobile GPUs in mainstream SoCs come in two architectures: ARM Mali series [5] and Qualcomm Adreno series [6], not NVIDIA products. These two series have just now incorporated hardware-level support for 8-bit integer multiplication units [5, 6]. Hence, many mobile GPU devices lacking hardware-level support for 8-bit integer operations still exist in the market.
> > >
> > > Furthermore, mainstream mobile inference engines are deficient in providing genuine support for 8-bit integer inference in the backend of mobile GPUs.
> > > Take the benchmarks of popular frameworks, TensorFlow Lite [7] and MNN [8], as examples.
> > > According to their documentation, these engines preemptively convert operators to their floating-point counterparts and invoke the floating-point backend for inference. Meanwhile, ARM Compute Library [9] also lacks the construction of 8-bit computation cores and resorts to converting quantized data to 32-bit lengths for backend computations. Note that the ARM Compute Library not only serves as a mainstream computing library for mobile devices, but it also supports multiple edge processors, e.g., the Raspberry Pi series. In summary, mainstream inference engines for mobile devices lack a comprehensive framework and built-in support for 8-bit integer quantized inference on mobile GPUs.
> > >
> > >
> > > [3] TensorRT. 2023. NVIDIA Deep Learning TensorRT Documentation. https://docs.nvidia.com/deeplearning/tensorrt/developer-guide/index.html
> > >
> > > [4] NVIDIA Architecture. 2023. https://www.nvidia.com/en-us/technologies/
> > >
> > > [5] Mali Graphics Processors. 2023. Arm GPUs for Graphics Processing. https://www.arm.com/products/silicon-ip-multimedia/gpu/mali-g720
> > >
> > > [6] Qualcomm Adreno. 2023. Adreno Graphics Processing Units. https://www.qualcomm.com/products/features/adreno
> > >
> > > [7] TensorFlow Lite. 2023. Deploy machine learning models on mobile and edge devices. https://www.tensorflow.org/lite
> > >
> > > [8] MNN. 2023. Alibaba MNN. https://github.com/alibaba/MNN
> > >
> > > [9] ARM, Available: https://arm-software.github.io/ComputeLibrary/v22.05.

---

> > > > ### Comment · Reviewer_ktdW · 2023-08-18
> > > >
> > > > Thanks for clarifying on the table.
> > > >
> > > > The mobile GPU story is still unclear to me, NVIDIA Jetson Orin seems to be an edge GPU to me, I do not fully understand the argument of "mobile GPU". If you really want to talk about the mobile story, Apple Neural Engine also support low-precision fixed-point (eg. 8-bit), the whole mindset of sticking with Qualcomm/ARM architecture seems unsupported to me.
> > > >
> > > > However, putting that aside, I think this paper does have enough contribution and have bumped my socre.

---

### Official Review · Reviewer_QVMx · 2023-07-06

**Soundness:** 3 good
**Presentation:** 2 fair
**Contribution:** 3 good
**Rating:** 7
**Confidence:** 5

**Summary:**

This paper introduces an innovative framework to optimize the implementation of Vision Transformers (ViTs) on mobile devices using sub-8-bit quantization. This framework identifies and addresses key challenges such as long-tailed distribution and systematic channel-wise outliers within ViT dataflow. Through the use of a three-step outlier-aware training process and an integer-only computation flow, PackQViT enhances the performance of ViTs. Additionally, a SIMD-based 4-bit packed multiplier was developed for efficient ViT acceleration. Compared to previous studies, PackQViT shows significant improvements in accuracy on the ImageNet dataset and speedups on a mobile implementation, marking a substantial advancement in the deployment of ViTs in resource-constrained environments.


**Strengths:**

This work directly addresses the numerous challenges faced by the application of ViTs on mobile devices and provides very practical solutions. More importantly, the authors have deployed and tested the algorithm mentioned in this paper on real hardware devices, which makes the paper more practical. Although they don't tell a very fancy story, this pragmatic style is exactly what the field of model compression and quantization needs today.

- The paper identifies and tackles key challenges within the ViT dataflow, including long-tailed distribution and systematic channel-wise outliers, making it effective for real-world deployments.
- The paper demonstrates a practical implementation of full sub-8-bit quantization for ViTs, a promising yet challenging technique for improving hardware efficiency. It also employs an innovative SIMD-based 4-bit packed multiplier for better end-to-end acceleration.
- The proposed three-step outlier-aware training approach can predict and regularize outliers, reducing complex hardware control logic during runtime inference.

**Weaknesses:**

The effectiveness of PackQViT is indeed reliant on specific hardware configurations, particularly SIMD-based units. The evaluation of PackQViT was conducted on a Snapdragon 870 SoC CPU, and its performance may vary on different hardware architectures. The results were presented primarily for ImageNet dataset and might not generalize across all kinds of data and diverse application domains.


**Questions:**

The Outlier-Aware Training methodology merits a more comprehensive explanation to enhance clarity. Can various inputs X yield diverse outliers, as well as predicted channel indices i and Power-of-Two Ratios r? What strategies are employed to identify and manage these outliers for differing inputs? How to fix the index i and r obtained in step 2 if the i and r is input-dependent? Furthermore, what is the protocol if multiple outliers are detected within the same channel?


**Limitations:**

The evaluation of PackQViT solely on the Snapdragon 870 SoC CPU limits the generalizability of the findings to a specific hardware environment. To establish a stronger foundation, it would indeed be beneficial to test PackQViT on a broader range of hardware devices representing diverse hardware and software environments commonly encountered in real-world applications.

---

> ### Author Rebuttal · Authors · 2023-08-09
>
> **ReW1**: For more results on another task, e.g., object detection, please refer to Reviewer yEDd: ReW3 which can validate that our proposed method can be generalized to other image domains. 2. Our 4-bit packed multiplier can be executed on mainstream processors on edge platforms (e.g., mobile phones, Raspberry Pis, and RISC-V IoT processor), which face challenges in processing low-bit data since their SIMD instructions only support 8-bit or wider data granularity. Hence, we test the speedup gain on multiple edge devices as shown in Table D. Note that INT8 matrix multiplication within PackQViT utilizes the original byte-level quantized GeMM kernel. For the 8-bit configuration, models can be speedup by up to 3.7x and up to 2.7x for DeiT and Swin models, respectively. For the 4-bit configuration,  models can be speedup by up to 5.5x and up to 3.5x for DeiT and Swin models, through our proposed packed 4-bit multiplier.  Thanks for your valuable suggestions. This one is constructive in enhancing our paper's generalization. We will add these results to the appendix of our final paper version.
>
> **Table D.** Hardware Results under Different Precision for Various ViTs on Multiple Edge Devices
> | Model | Method           | #Bits | Size(MB) | Android CPU (s) | RaspberryPi (s) | RISC-V (s)|
> |-----------|-------------------|-----|-----|-----|------|-----|
> | DeiT-T | Full-precision | 32 | 20.1 | 0.172 | 23.97 | 3.4   |
> | DeiT-T |PackQViT         | 8   | 5.2   | 0.056 | 7.73   | 1.11 |
> | DeiT-T | PackQViT      | 4   | 2.7   | 0.037 | 5.15   | 0.74 |
> | DeiT-S | Full-precision | 32 | 88.2 | 0.363 | 59.75 | 7.2   |
> | DeiT-S | PackQViT        | 8   | 22.2 | 0.110 | 17.5   | 2.2   |
> | DeiT-S | PackQViT      | 4   | 11.4 | 0.073 | 11.97 | 1.46 |
> | DeiT-B | Full-precision | 32 | 346.2 | 0.858 | 143.4 | 17   |
> | DeiT-B | PackQViT       | 8   | 86.8 | 0.233 | 38.24   | 4.6   |
> | DeiT-B | PackQViT      | 4   | 44.1 | 0.156 | 26.12 | 3.16 |
> | Swin-T | Full-precision | 32 | 114.2 | 0.276 | 44.9 | 5.5  |
> | Swin-T | PackQViT        | 8   | 28.6 | 0.099 | 16.4   | 2.1  |
> | Swin-T | PackQViT      | 4   | 14.6 | 0.085 | 13.76 | 1.69 |
> | Swin-S | Full-precision | 32 | 199.8 | 0.55 | 89.78 | 10.9  |
> | Swin-S | PackQViT       | 8   | 50.2 | 0.213 | 34.4   | 4.2  |
> | Swin-S | PackQViT      | 4   | 25.5 | 0.164 | 25.93 | 3.29 |
>
> **ReQ1**: We show one structured pattern of channel-wise outliers: most of the outliers (over 91.5%) appear in the fixed channels of the fixed block. To validate this structured/fixed pattern, we run multiple models from small sizes (DeiT-T/Swin-T) to large models (DeiT-B/Swin-B) on ImageNet-1k, MS COCO, and Cifar-100. This similar phenomenon is found in [8] for Bert models. We performed a similar analysis on ViTs and verified this law's applicability to ViTs. Even though some outliers are not in the predicted channels, the related quantization error can be eliminated by our training process. Furthermore, suppose multiple outliers are detected within the same channel. In that case, we still use the power-of-two coefficient to refine the scaling factor of that channel, which regularizes the data distribution into a smaller range that is similar to others. After that, we can perform layer-wise quantization on the whole activation matrix. Please refer to Reviewer 7eJH: Q2 for more hardware implementation details for this trick.
>
> **ReLimitation.**: Thanks for your valuable suggestions which can validate the generalization of the proposed packed 4-bit multiplier. We also test the inference implementation on various edge devices, including the RealmeGT Android Phone with Snapdragon 870 SoC, Raspberry4 B with Quad-core CPU and the simulator for RISC-V ultra-low power IoT processor. Please refer to ReW1 for detailed results.
>
> [8] Dettmers, Tim, et al. "Llm. int8 (): 8-bit matrix multiplication for transformers at scale." Neurips, 2022.

---

> > ### Comment · Area_Chair_brkv · 2023-08-18
> >
> > Dear reviewer,
> >
> > The discussion with authors is closing soon, please review the rebuttal to see if the authors have addressed your concerns, and acknowledge to the authors that you have read their response.
> >
> > Thanks

---

> > ### Comment · Reviewer_QVMx · 2023-08-22
> >
> > Dear authors, most of my concerns have been addressed. Thank you for the detailed experiments on various platforms. I'll maintain my score.

---

### Official Review · Reviewer_7eJH · 2023-07-26

**Soundness:** 3 good
**Presentation:** 3 good
**Contribution:** 3 good
**Rating:** 5
**Confidence:** 5

**Summary:**

This work started from analyzing the distribution of weights and activations of ViT models, identified several characteristics of critical layers, and applied Integer and hardware friendly techniques to address the issues. Furthermore, the author demonstrated a fully quantized ViT (including LayerNorm, Softmax, and GELU) and did a nice profiling works on mobile CPU to verify the benefits of INT8 and INT4 models. Specifically, INT4 instructions are not natively supported on this HW platform hence required a custom kernel based on INT8 instruction sets. Eventually, the author achieved 3-6x speed-up over FP32 model with comparable or slightly higher accuracy on ImageNet-1k.

**Strengths:**

Originality:
Marginal
Most of the techniques employed in this work seems to have been reported in Ref 31, FQ-ViT, already. The new components are distribution analysis, outlier-aware training, the implementation of specialized HW kernels, and using profiling to demo the importance of quantizing Softmax/LayerNorm/GELU in addition to matmuls on mobile CPU. The reviewer would consider this as "application to a new domain" compared to Ref 31.

Quality:
Good

Clarity:
Good

Significance:
Could be a good reference for quantization on mobile devices.

**Weaknesses:**

1. Novelty, as mentioned above.

2. A few claims/explanations may not be well-supported, for example:
**Line 292**, *"(QAT having higher accuracy than FP baseline) suggests that redundancy in the original model hinders it from converging to the optimum"*

Because the QAT in this work is a 70-epochs fine-tuning on top of the baseline FP32, the accuracy improvement could have been derived from this fine-tuning instead of the quantization itself. In fact, this behavior is not uncommon in QAT/fine-tuning. On the other hand, to support the "explanation", the author would need to "fine-tune" the FP32 with the same settings and compare it with INT8 results.

In **section 5.3**, take INT4 cases as an example, the author seems to compare accuracy differences within a small range based on a single run or best record. It would be more reasonable if the author could report a 5-run average of the baseline or FP32 model. After taking run-to-run variations into account, author could have claimed some options "comparable" instead of "suffer a performance drop by 0.1%" as on **Line 323.**

3. Although it's obvious for readers with background knowledge, it would be better to define variables like z and s in Eq 1, as these are used in later equations as well. Also a couple typos, like Eq 5 and third row in the blue block on Fig. 5, .

**Questions:**

1. The choice of log2-attention seems to be a bit counter-intuitive. As the attention map is generally understood as "highlighting the importance of the content", why would adding more granularities to the "unimportant part", i.e. close to 0 attention score, help the model accuracy? In fact, if this technique helps as the author claims, we would expect it to be more apparent on INT4 rather than INT8 on Table 3, because effective utilization of the limited bins should be more critical for INT4.

2. Around **Line 186**, author compares the way channel-wise outliers are handled to "online data-adaptive selection", (which is sometimes called dynamic quantization). But this technique as described in **Sec 4.3** would be closer to a "per-channel quantization for activations." And the per-channel option might be more generalizable than the technique described here. Maybe the author could comment on the differences and requirement for HW/SW implementation between the reported technique vs per-channel quantization for activations.

**Limitations:**

No societal impact.

---

> ### Author Rebuttal · Authors · 2023-08-09
>
> **ReW1**: Our difference between Ref [31] can be categorized into four aspects, including motivations, optimization of task accuracy, optimization of hardware efficiency, and evaluation results.
>
> Motivation: We analyze data distribution within ViTs (long-tail distribution and outlier analysis), which provides insights on quantizing model weights and activations effectively to enhance task accuracy and on-device efficiency. Meanwhile, we conduct a hardware latency profiling to figure out the runtime bottleneck of the whole models, which guides us to optimize operator kernels for efficiency.
>
> Optimization of task accuracy: Different from Ref [31] which uses an input-adaptive way to runtime detect and optimize outliers during model inference. We present one structured pattern of channel-wise outliers: most outliers (over 91.5%) appear in the fixed channels of the fixed block. So we predict outliers' channel index and power-of-two coefficients and mitigate the adverse effects of outliers through outlier-aware training, not dynamic inference. To validate this structured/fixed pattern, we run multiple models from small sizes (DeiT-T/Swin-T) to large models (DeiT-B/Swin-B) on ImageNet-1k, MS COCO, and Cifar-100.
>
> Optimization of hardware efficiency: (1) We figure out the nonlinear operators as the speed bottleneck. Then, we adapt them into hardware-friendly ones differently from Ref [31]. Int-2n-Softmax: We replace the exponent e with a faster operation,  power-of-2 to bring more efficiency benefits without accuracy degradation;  GeLU: Based on [5], we adapt it with efficient hardware implementation (detailed implementation in Reviewer yEDd: ReW2 and Reviewer ysuz: ReW1). We have mentioned these in our appendix and will add more details in our final version. (2) Ref [31] uses a runtime dynamic selection method to optimize outliers, while we leverage structured/fixed pattern of data outliers to predict their channel index and power-of-two coefficients in advance through training. Then we can preload these indexes and coefficients with the model weight and compute them simultaneously with the matrix multiplication between weights and activations. Thus, our design can avoid dynamic overhead which is significantly huge in the practical implementation. Also, the dynamic processing on activations is not well-supported in the common inference engine, such as ArmComputeLibrary. (3) We quantize models into lower bits 4-bit. To achieve practical acceleration, we propose the packed 4-bit multiplier to support this sub-8-bit matrix multiplication, as mentioned in  Reviewer yEDd: W2.
>
> Evaluation results: We not only can get better accuracy results than Ref [31], but also accelerate end-to-end on-device latency performance of the quantized ViTs on multiple edge devices (Results in Reviewer QVMx: ReW1).
>
> **ReW2**: (1) Following your suggestion, we add a group of experiments that only fine-tune models under the same setting (100 epochs & distillation) without quantization, as shown in Table C.
>
> **Table C.**
> |Model   | Fine Tuning   | #Bits | Top-1 Acc.(%)|
> |-----------|------------------|-----|-----|
> | DeiT-T | N (baseline)  | 32 | 72.2   |
> | DeiT-T | Y                   | 32 | 74.1   |
> | DeiT-T | Y                   | 8   | 74.5   |
> | Swin-T | N (baseline)  | 32 | 81.35 |
> | Swin-T | Y                   | 32 | 82      |
> | Swin-T | Y                   | 8   | 82.3   |
>
> With pure finetuning, the accuracy of full-precision models can be increased by 1.9% and 0.65% for DeiT-T and Swin-T, respectively. Combined with quantization, the accuracy can be further increased by 0.4% and 0.3% for DeiT-T and Swin-T. This illustrates that quantization-aware training to 8-bit can contribute to redundancy reduction to enhance task accuracy (the same as [6]). Even though the finetuning can help the accuracy, our method deploys the same distillation as Q-ViT and ends with better task accuracy than it.
>
> (2) We have run the Q-ViT baseline 5 more times these days. We get the average result of 0.15% with a standard deviation = 1.0724. We will add these details in Line 323.
>
> **ReW3**: We'll fix these typos in our final version.
>
> **ReQ1**: 1. Remaining unimportant tokens' information still helps improve task accuracy [7]. We cannot use the token packing technique mentioned in [7] which averages the unimportant tokens into one package token to recover the information within unimportant tokens. Still, we can mitigate the degree of aggressive quantization. 2. In Table 3 of our paper, the accuracy increase is 0.5% for INT8 and 0.3% for INT4. Information degradation is so much in 4-bit, which will mitigate the recoverability of 4-bit.
>
> **ReQ2**: Our method is different from per-channel quantization for outliers. We perform layer-wise quantization on the activation matrix; all the channels share the same quantization parameters, i.e., scaling factors and zero-point. However, the predicted power-of-two coefficients will refine the outlier channels' scaling factors. In the practical implementation, those power-of-two coefficients will be equivalently mathematically transformed over the corresponding weights, as shown in Figure A.
>
> Our framework is compatible with per-channel quantization. However, common inference engines usually provide support only for layer-wise quantization on activation and per-channel quantization on weights, such as the Gemm and Convolution operator configuration. For example, ArmComputeLibrary[1] only supports channel-wise quantization configuration for weight matrix, instead of the input activation.
>
> These questions can help us enrich the content of the paper. We will add these details to our final version.
>
> [5] Sehoon, et al. "I-bert: Integer-only bert quantization", ICML, 2021.
>
> [6] Liang, et al. "Pruning and quantization for deep neural network acceleration: A survey." Neurocomputing, 2021.
>
> [7] Kong, et al. "Spvit: Enabling faster vision transformers via latency-aware soft token prunin," ECCV, 2022.

---

> > ### Comment · Reviewer_7eJH · 2023-08-18
> >
> > Thanks for the detailed explanations. The reviewer has no further comments and no needs to change the ratings.

---

### Official Review · Reviewer_yEDd · 2023-07-27

**Soundness:** 4 excellent
**Presentation:** 3 good
**Contribution:** 3 good
**Rating:** 6
**Confidence:** 4

**Summary:**

In this paper, the authors proposed PackQViT, a fast and accurate sub-8-bit quantization for running ViT on mobile devices. To maintain accuracy, the authors proposed different quantization schemes for activation and weights, and an outlier-aware training stage. For efficiency, they proposed fully integer softmax and layernorm, and a SIMD-based 4-bit packed multiplier. Overall, the method can accelerate the inference of ViT models on mobile devices at a small accuracy degradation.


**Strengths:**

1. Firstly, the paper is generally well written and easy to follow.
2. The authors present a carefully designed algorithm & system co-design solution to improve the accuracy and efficiency of the ViT models. The observation on the different distribution of models weights and activations are quite insightful.
3. The results are very solid, with both high accuracy numbers and actual speed up under 8-bit/4-bit cases.

**Weaknesses:**

1. The accuracy numbers in Table 1 and 3 are a bit confusing: in many cases, the 8-bit accuracy is higher than the original FP32 model, which makes the ablation study hard to compare. Does it indicate the original model is not well trained? It is more proper to use a model trained with a good schedule, instead of undertrained ones.
2. The paper lacks a detailed discussion on the kernel implementation. There are a lot of different workloads like integer quantization, log quantization, packed 4-bit multiplication, etc. It would be very helpful to give more discussion on how to implement the kernel so that it runs efficiently on smartphone SoC.
3, It would be more convincing if the authors can also add some results on other tasks apart from image classification, like detection, for example.

**Questions:**

Please see above 1 and 2 in the weakness part.

**Limitations:**

Yes

---

> ### Author Rebuttal · Authors · 2023-08-09
>
> **ReW1**: We appreciate your valuable comments since this allows us to clarify this phenomenon.
> The accuracy of the full precision model in Table 1 of our paper is sourced from the original paper of DeiT and Swin, both of which do not perform the DGD distillation techniques. Q-ViT proposes DGD distillation, and our training setting uses this distillation technique, as mentioned in Line 270. This is why we can get better accuracy results than the full precision models. It is worth mentioning that we can achieve significantly better task accuracy with the same training setting than Q-ViT.
>
> **Table A.** Comparison of Full Precision models w/ & w/o DGD Distillation
> |Model| DGD Distillation | #Bits | Top-1 Acc.(%)|
> |------------|-----|-----|-----|
> |DeiT-T|N    |32  |72.2	|
> |DeiT-T|Y    |32 |74.4	|
> |Swin-T|N    |32  |81.35	|
> |Swin-T|Y    |32. |82.2	|
>
> Following your valuable suggestions, we also train the full precision model from scratch with the DGD distillation techniques, as shown in Table A. It shows that the model accuracy can be further increased by 2.2% and 0.85% for DeiT-T and Swin-T, respectively. For more analysis of the distillation techniques, please refer to Reviewer 7eJH: W2.
>
> **ReW2**: We appreciate your valuable comments since this allows us to describe our hardware deployment in detail.
> The implementation below has been programmed based on ArmISAs and tested on the maincore 3.2GHz Cortex A77 of Snapdragon 870 onboard CPU. The kernels have been implemented within ArmComputeLibrary v22.05 [1] Inference framework.
>
> (1) Integer Quantization Computation Flow in current inference engines:
> 	QuantizeLayer(Scale, Zeropoint) -> Quantized Operators -> DequantizeLayer (Scale, Zeropoint) with the data type conversion sequence of: FP32 -> INT -> FP32 -> …, thus the inter-operator data transmission is still in floating-point format, ensuring the mathematical correctness with scale & zeropoint of the current quantization inference.
>
> (2) Softmax Implementation based on Log2Quantization:
>
> 	Log2Softmax(in_ptr, tmp_ptr, length):
>     sum = 0
>     vec_max = Globalmax
>     /* 2^n & Reduced Sum */
>     for i = 0 to length, step 16:
>         in = load(in_ptr + i)
>         in = vec_max - in
>         in = 1 << in;
>         sum = sum + sum_vec(in)
>         store(tmp_ptr + i, in)
>     /* Normalize Log2SFM */
>     for i = 0 to length, step 16:
>         temp = load(tmp_ptr + i)
>         res = log2(Approximate_nearest_POW2(sum / temp))
>         res = M - res
>
> Compared with the computation of exp(x) in common softmax inference, where the approximation is usually calculated via 3 sequential steps: Range Reduction, Polynomial Approximation, and Reconstruction, which requires more than 13 floating-point operation instructions for a single function implementation (e.g., sfm_impl.cpp on ArmComputeLibrary), our flow removes the costly floating-point calculations together with reducing memory allocation for temporal result storage. Via integer-based LeftShift operations and parallel processing for 16 operands, efficient Softmax with preserved accuracy and computational savings could be achieved.
>
> (3) Packed 4-bit Multiplier:
> 	The breakdown of MLA (multiplication & Addition) with MUL and ADD does not introduce any obvious inference latency differences according to our benchmarks on both Snapdragon 870 onboard-CPU and RaspberryPi 4B, opening up the possibility of inserting result-adjustment auxiliary operations between the MUL and ADD to achieve SMMW (single-multiplication-multiple-weight).
> The 4-bit Multiplier has been implemented based on ARM ISAs following:
> 	MUL (Multiplication), LSL (LeftShift), ORR (BitwiseOR), AND (BitwiseAND) the same process as Figure 5 in our paper.
>
> 	4-bit_GeMM_4x4(i, s1, s2):
>     s1 = [ 4x4 matrix of src1 ]
>     s2 = [ 1x4 vector of src2 ]
>     c = [ 4x4 matrix of zeros ]
>     mask = [ 1x4 vector of 0x00FF00FF ]
>     p = [ 1x4 vector with zeros ]
>     /* Inner loop for 16 elements
>        4 units per loop */
>     for j in range(4):
>         /* Lane-wise Multiplication */
>         p = [multiply the j-th row of s1 with the i-th element of s2]
>         /* Product Rearrangement */
>         t = [left shift elements of p by 8 bits]
>         p = [bitwise OR between p, t]
>         p = [bitwise AND between p, mask]
>         /* Accumulation */
>         c.row[j] = [add elements of p to the corresponding j-th row of c]
>     return c
> **ReW3**:  We further compare object detection with transformers (DETR) [2], an end-to-end detector via a transformer encoder-decoder. We perform our quantization on DETR-R50. We compare the large-scale COCO [3] dataset as shown in Table B. We compare our method under the 4-bit precision. We also report the detection performance of the 8-bit PTQ method, VT-PTQ [4].
>
> **Table B.** Evaluation Results with our proposed method using DETR-50 on COCO val2017. #Bits denotes the bit-width of weights, activations, and attention activations are quantized into #Bits precision.
> |Methods| #Bits |mAP | AP50| AP75|
> |------------|-----|-----|-----|-----|
> |full-precision|32  |	59.5|	83.3|	64.7|
> |VT-PTQ       |8    |	57.6|	82.3|	63.1|
> |Ours             |8    |	58.3|	82.9|	63.9|
> |Ours		|4    |	55.6|	82.2|	60|
>
> For this more complex task, our method has a negligible detection performance degradation by 1.2% mAP, which is better than the VT-PTQ method by 0.7 mAP. For more aggressive quantization under 4-bit, the model has 3.9 mAP performance drops. For more complex tasks, quantization tends to lead to a degradation of task performance.
>
> Thanks again for your suggestions to improve our paper presentation. We will add these details in our final version.
>
> [1] ARM, Available: https://arm-software.github.io/ComputeLibrary/v22.05.
>
> [2] Carion,  et al. "End-to-end object detection with transformers" ECCV, 2020.
>
> [3] Lin, et al. "Microsoft coco: Common objects in context", ECCV, 2014.
>
> [4] Liu,  et al. "Post-training quantization for vision transformer", Neurips, 2021.

---

> > ### Comment · Reviewer_yEDd · 2023-08-17
> > **Updated review**
> >
> > Thank the authors for the helpful discussion, which has largely addressed my concerns. I have raised my rating from 5 to 6.

---

### Author Rebuttal · Authors · 2023-08-09

We first sincerely thank every reviewer for your insightful and constructive feedback. Then, we will answer the specific questions from each reviewer. We upload a pdf file with figures (Figure A,B,C) and equations (Equation 1,2) that we will present in our rebuttal.

---

### Decision · Program_Chairs · 2023-09-21

**Decision:**

Accept (poster)

**Comment:**

This paper presents a framework for quantization vision transformer (ViT) models on both linear and non-linear operations allowing their deployment on edge devices. The paper applies various quantization schemes based on the data-distribution analysis to preserve model accuracy. Additionally, custom kernels are implemented, showing speedup on edge CPUs.

The paper is in general well-written. The insights provided through data distribution analysis and the latency breakdown are particularly enlightening. The introduced framework is practical. The evaluation is comprehensive, demonstrating model accuracy across diverse tasks and acceleration on various edge devices.

During the rebuttal phase, the authors conscientiously responded to queries and concerns, providing additional data and detailed explanation on kernel implementation, object detection task results, and latency on diverse edge devices. The effort has led to improvement on reviewer ratings. Those additional materials should be integrated into the final version of the paper.

The main weakness of the paper is its relatively limited novelty, particularly regarding the quantization techniques which seems to draw from existing methods. However, considering this framework to be valuable in practical as a reference for the community seeking to deploy ViT models on edge devices, the paper is recommended for acceptance.